# Payoff-based Learning with Matrix Multiplicative Weights in Quantum Games

**Kyriakos Lotidis**
Stanford University
klotidis@stanford.edu

**Panayotis Mertikopoulos**
Univ. Grenoble Alpes, CNRS, Inria, Grenoble INP
LIG 38000 Grenoble, France
panayotis.mertikopoulos@imag.fr

**Nicholas Bambos**
Stanford University
bambos@stanford.edu

**Jose Blanchet**
Stanford University
jose.blanchet@stanford.edu

## Abstract

In this paper, we study the problem of learning in quantum games – and other classes of semidefinite games – with scalar, payoff-based feedback. For concreteness, we focus on the widely used *matrix multiplicative weights* (MMW) algorithm and, instead of requiring players to have full knowledge of the game (and/or each other's chosen states), we introduce a suite of *minimal-information matrix multiplicative weights* (3MW) methods tailored to different information frameworks. The main difficulty to attaining convergence in this setting is that, in contrast to classical finite games, quantum games have an infinite continuum of pure states (the quantum equivalent of pure strategies), so standard importance-weighting techniques for estimating payoff vectors cannot be employed. Instead, we borrow ideas from bandit convex optimization and we design a zeroth-order gradient sampler adapted to the semidefinite geometry of the problem at hand. As a first result, we show that the 3MW method with deterministic payoff feedback retains the $\mathcal{O}(1/\sqrt{T})$ convergence rate of the vanilla, full information MMW algorithm in quantum min-max games, even though the players only observe a single scalar. Subsequently, we relax the algorithm's information requirements even further and we provide a 3MW method that only requires players to observe a random realization of their payoff observable, and converges to equilibrium at an $\mathcal{O}(T^{-1/4})$ rate. Finally, going beyond zero-sum games, we show that a regularized variant of the proposed 3MW method guarantees local convergence with high probability to all equilibria that satisfy a certain first-order stability condition.

## 1 Introduction

The integration of quantum information theory into computer science and machine learning [4, 51, 64] has the potential ofy providing faster and more efficient computing resources, new encryption and security protocols, and improved machine learning algorithms, enabling advancements in areas such as quantum cryptography, shadow tomography, quantum GANs, and adversarial learning [1, 15, 18, 37]. As a well-known example, Google's "Sycamore" 54-qubit processor recently showcased this "quantum advantage" by training an autonomous vehicle model in less than 200 seconds [4], a fact made possible by the ability of quantum computers to prepare superpositions of qubits that exceed the operational capabilities of standard Boolean gates.

Deploying such models within a multi-agent context, such as the utilization of QGANs or autonomous vehicles, leads to a significant transformation compared to classical non-cooperative environments.

37th Conference on Neural Information Processing Systems (NeurIPS 2023).

Indeed, unlike classical games (where a mixed strategy is a probabilistic mixture of the underlying pure strategies), quantum games utilize mixed states, which represent probabilistic mixtures of quantum projectors. As a consequence, a mixed quantum state can yield payoffs that cannot be expressed as a convex combination of classical pure strategies.

In light of this, quantum learning has drawn significant attention in recent years [2, 27, 28, 33, 34, 60]. In a multi-agent context, the most widely used framework is the so-called *matrix multiplicative weights* (MMW) algorithm [2, 16, 27, 28, 35]: First introduced by Tsuda et al. [59] in the context of matrix and dictionary learning, MMW can be viewed as a semidefinite analogue of the standard Hedge / EXP3 methods for multi-armed bandits [6, 36, 61], and is a special case of the mirror descent family of algorithms [47]. Specifically, in the contrete setting of two-player, zero-sum quantum games, Jain & Watrous [27] showed that players using the MMW algorithm can learn an $\varepsilon$-equilibrium in $\mathcal{O}(1/\varepsilon^2)$ iterations – or, in terms of speed of convergence after $T$ iterations, they converge to equilibrium at a $\mathcal{O}(1/\sqrt{T})$ rate.

To the best of our knowledge, this result remains the tightest known bound for equilibrium learning in quantum games – and the more general class of semidefinite games [26]. At this point, we highlight that we focus on classical computing algorithms for solving quantum games, unlike recent results [10, 22] that employ quantum algorithms to solve classical games and semidefinite programs. Building on [27], Jain et al. [28] studied its continuous-time analogue – the *quantum replicator dynamics* (QRD) – in quantum min-max games, focusing on the recurrence and volume conservation properties of the players' actual trajectory of play. Going beyond the min-max case, [38] examined the convergence of the dynamics of *"follow the quantum leader"* (FTQL), a class of continuous-time dynamics that includes the QRD as a special case. The main result of [38] was that the only states that are asymptotically stable under the (continuous-time) dynamics of FTQL are those that satisfy a certain first-order stationarity condition known as *variational stability* [44, 46]. In a similar line of work, Lin et al. [35] studied the continuous-time QRD, and discrete-time MMW in quantum potential games, utilizing a Riemannian metric to obtain a gradient flow in the spirit of [41, 42].

**Our contributions in the context of previous work.** All works mentioned above, in both continuous and discrete time, assume full information, i.e., players have access to their individual payoff gradients – which, among others, might imply that they have full knowledge of the game. However, this condition is rarely met in online learning environments where players only observe their in-game payoffs; this is precisely the starting point of our paper which aims to derive a convergent payoff-based, gradient-free variant of MMW algorithm for learning in quantum games.

A major roadblock in this is that standard approaches from learning in finite games fail in the quantum setup for two reasons: First and foremost, there is a *continuum* of pure states available to every player, unlike classical finite games where there is only a *finite* set of pure actions. Second, even after the realization of the pure states of the players, there is an inherent uncertainty and randomness due to the payoff-generating quantum process (an aspect that has no classical counterpart). To overcome this hurdle, we employ a continuous-action reformulation of quantum games, and we leverage techniques from bandit convex optimization for estimating the players' payoff gradients.

Our first contribution is a variant of MMW that only requires mixed payoff observations and achieves an $\mathcal{O}(1/\sqrt{T})$ equilibrium convergence rate in two-player zero-sum quantum games, matching the rate of the full information MMW in [27]. Then, to account for information-starved environments where players are only able to observe their in-game, realized payoff observable, we also develop a bandit variant of MMW which utilizes a single-point gradient estimation technique in the spirit of [55] and achieves an $\mathcal{O}(T^{-1/4})$ equilibrium convergence rate. Finally, we also examine the behavior of the MMW algorithm with bandit information in general $N$-player games, where we show that variationally stable equilibria are locally attracting with high probability.

Importantly, the above results transfer to more general games with a semidefinite structure – such as multi-agent covariance matrix optimization in signal processing, energy efficiency maximization in multi-antenna systems, etc. [43, 45, 62]. While we do not provide a complete theory, we discuss a number of non-quantum applications that showcase how our results can be generalized further.

**Notation.** Given a (complex) Hilbert space $\mathcal{H}$, we will use Dirac's bra-ket notation and write $|\psi\rangle$ for an element of $\mathcal{H}$ and $\langle\psi|$ for its adjoint; otherwise, when a specific basis is implied by the context, we will use the dagger notation "†" to denote the Hermitian transpose $\psi^\dagger$ of $\psi$. We will also write

$\mathbb{H}^d$ for the space of $d \times d$ Hermitian matrices, and $\mathbb{H}^d_+$ for the cone of positive-semidefinite matrices in $\mathbb{H}^d$. Finally, we denote by $\|\mathbf{A}\|_F = \sqrt{\operatorname{tr}[\mathbf{A}^\dagger \mathbf{A}]}$ the Frobenius norm of $\mathbf{A}$ in $\mathbb{H}^d$.

## 2 Problem setup and preliminaries

We begin by reviewing some basic notions from the theory of quantum games, mainly intended to set notation and terminology; for a comprehensive introduction, see [23]. To streamline our presentation, we introduce the primitives of quantum games in a 2-player setting before treating the general case.

**Quantum games.** Following [20, 23], a 2-player *quantum game* consists of the following:

1. Each player $i \in \mathcal{N} \coloneqq \{1, 2\}$ has access to a complex Hilbert space $\mathcal{H}_i \cong \mathbb{C}^{d_i}$ describing the set of (pure) *quantum states* available to the player (typically a discrete register of qubits). A quantum state is an element $\psi_i$ of $\mathcal{H}_i$ with unit norm, so the set of pure states is the unit sphere $\Psi_i \coloneqq \{\psi_i \in \mathcal{H}_i : \|\psi_i\|_F = 1\}$ of $\mathcal{H}_i$. We will write $\Psi \coloneqq \Psi_1 \times \Psi_2$ for the space of all ensembles $\psi = (\psi_1, \psi_2)$ of pure states $\psi_i \in \Psi_i$ that are independently prepared by each player.
2. The rewards that players receive are based on their individual *payoff functions* $u_i \colon \Psi \to \mathbb{R}$, and they are derived through a *positive operator-valued measure* (POVM) quantum measurement process. Following [17], this unfolds as follows: Given a *finite* set of *measurement outcomes* $\Omega$ that a referee can observe from the players' quantum states (e.g., measure a player-prepared qubit to be "up" or "down"), each outcome $\omega \in \Omega$ is associated to a positive semi-definite operator $\mathbf{P}_\omega \colon \mathcal{H} \to \mathcal{H}$ defined on the tensor product $\mathcal{H} \coloneqq \mathcal{H}_1 \otimes \mathcal{H}_2$ of the players' individual state spaces. We further assume that $\sum_{\omega \in \Omega} \mathbf{P}_\omega = \mathbf{I}$ so the probability of observing $\omega \in \Omega$ at state $\psi \in \Psi$ is $P_\omega(\psi) = \langle \psi_1 \otimes \psi_2 | \mathbf{P}_\omega | \psi_1 \otimes \psi_2 \rangle$.
3. The payoff of each player is then generated by this measurement process via a *payoff observable* $U_i \colon \Omega \to \mathbb{R}$: specifically, the measurement $\omega$ is drawn from $\Omega$ based on the players' state profile $\psi = (\psi_1, \psi_2)$, and each player $i \in \mathcal{N}$ receives as reward the quantity $U_i(\omega)$. Accordingly, the player's expected payoff at state $\psi \in \Psi$ is $u_i(\psi) \coloneqq \langle U_i \rangle \equiv \sum_\omega P_\omega(\psi) U_i(\omega)$.

A *quantum game* is then defined as a tuple $\mathcal{Q} \equiv \mathcal{Q}(\mathcal{N}, \Psi, u)$ with players, states, and payoff as above.

**Mixed states.** Apart from pure states, each player $i \in \mathcal{N}$ may prepare probabilistic mixtures thereof, known as *mixed states*. These mixed states differ from mixed strategies used in classical, finite games as they do not correspond to convex combinations of their pure counterparts; instead, given a family of pure quantum states $\psi_{i\alpha_i} \in \Psi_i$ indexed by $\alpha_i \in \mathcal{A}_i$, a mixed state is described by a *density matrix* of the form

$$\mathbf{X}_i = \sum_{\alpha_i \in \mathcal{A}_i} x_{i\alpha_i} |\psi_{i\alpha_i}\rangle \langle \psi_{i\alpha_i}| \tag{1}$$

where the *mixing weights* $x_{i\alpha_i} \geq 0$ of each $\psi_{i\alpha_i}$ are normalized so that $\operatorname{tr} \mathbf{X}_i = 1$. By Born's rule, this means that the probability of observing $\omega \in \Omega$ under $\mathbf{X} = (\mathbf{X}_1, \mathbf{X}_2)$ is

$$P_\omega(\mathbf{X}) = \sum_{\alpha_1 \in \mathcal{A}_1} \sum_{\alpha_2 \in \mathcal{A}_2} x_{1,\alpha_1} x_{2,\alpha_2} P_\omega(\psi_\alpha). \tag{2}$$

where $\psi_\alpha = \psi_{1,\alpha_1} \otimes \psi_{2,\alpha_2}$. Therefore, in a slight abuse of notation, the expected payoff of player $i \in \mathcal{N}$ under $\mathbf{X}$ will be $u_i(\mathbf{X}) = \sum_{\alpha \in \mathcal{A}} x_\alpha u_i(\psi_\alpha)$. which, equivalently, can be written as:

$$u_i(\mathbf{X}) = \sum_{\omega \in \Omega} \sum_{\alpha \in \mathcal{A}} x_\alpha u_i(\psi_\alpha) U_i(\omega). \tag{3}$$

This gives a succint representation of the payoff structure of $\mathcal{Q}$ – see also Eq. (5) below.

**Continuous game reformulation.** In view of the above, treating a quantum game as a "tensorial" extension of a finite game can be misleading. For our purposes, it would be more suitable to treat a quantum game as a *continuous game* where each player $i \in \mathcal{N}$ controls a matrix variable $\mathbf{X}_i$ drawn from the "spectraplex" defined as $\mathcal{X}_i = \{\mathbf{X}_i \in \mathbb{H}^{d_i}_+ : \operatorname{tr} \mathbf{X}_i = 1\}$. In this interpretation, the players' payoff functions $u_i \colon \mathcal{X} \equiv \mathcal{X}_1 \times \mathcal{X}_2 \to \mathbb{R}$ are *linear* in each player's density matrix $\mathbf{X}_i \in \mathcal{X}_i, i \in \mathcal{N}$. Since $u_1, u_2$ are linear in $\mathbf{X}_1$ and $\mathbf{X}_2$, the individual payoff gradients of each player will be given by

$$\mathbf{V}_1(\mathbf{X}) \coloneqq \nabla_{\mathbf{X}_1^\top} u_1(\mathbf{X}) \quad \text{and} \quad \mathbf{V}_2(\mathbf{X}) \coloneqq \nabla_{\mathbf{X}_2^\top} u_2(\mathbf{X}) \tag{4}$$

so we can further write each player's payoff function as

$$u_1(\mathbf{X}) = \operatorname{tr}[\mathbf{X}_1 \mathbf{V}_1(\mathbf{X})] \quad \text{and} \quad u_2(\mathbf{X}) = \operatorname{tr}[\mathbf{X}_2 \mathbf{V}_2(\mathbf{X})] \quad \text{for all } \mathbf{X} \in \mathcal{X}. \tag{5}$$

Since $\mathcal{X}$ is compact and each $u_i$ is multilinear in $\mathbf{X}$, the players' payoff functions are automatically bounded, Lipschitz continuous and Lipschitz smooth, i.e., there exist constants $B_i$, $G_i$ and $L_i$, $i \in \mathcal{N}$, such that, for all $\mathbf{X}, \mathbf{X}' \in \mathcal{X}$, we have:

1. Boundedness: $\quad\quad\quad\;\; |u_i(\mathbf{X})| \leq B_i$
2. Lipschitz continuity: $\quad |u_i(\mathbf{X}) - u_i(\mathbf{X}')| \leq G_i \|\mathbf{X} - \mathbf{X}'\|_F$
3. Lipschitz smoothness: $\quad \|\mathbf{V}_i(\mathbf{X}) - \mathbf{V}_i(\mathbf{X}')\|_F \leq L_i \|\mathbf{X} - \mathbf{X}'\|_F$

**Nash equilibrium.** The most widely used solution concept in game theory is that of a *Nash equilibrium* (NE). In our context, it is mixed profile $\mathbf{X}^* \in \mathcal{X}$ from which no player has incentive to deviate, i.e., $u_1(\mathbf{X}^*) \geq u_1(\mathbf{X}_1; \mathbf{X}_2^*)$ and $u_2(\mathbf{X}^*) \geq u_2(\mathbf{X}_1^*; \mathbf{X}_2)$ for all $\mathbf{X}_1 \in \mathcal{X}_1, \mathbf{X}_2 \in \mathcal{X}_2$. Since $\mathcal{X}_i$ is convex and $u_i$ linear in $\mathbf{X}_i$, the existence of Nash equilibria follows from the Debreu's theorem [19].

**Zero-sum quantum games.** In the case where $u_1 = -u_2$, and setting $\mathcal{L} \colon \mathcal{X}_1 \times \mathcal{X}_2 \to \mathbb{R}$, the Nash equilibria of $\mathcal{Q}$ are the saddle points of $\mathcal{L}$, i.e., the solutions of the minimax problem

$$\max_{\mathbf{X}_1 \in \mathcal{X}_1} \min_{\mathbf{X}_2 \in \mathcal{X}_2} \mathcal{L}(\mathbf{X}_1, \mathbf{X}_2) = \min_{\mathbf{X}_2 \in \mathcal{X}_2} \max_{\mathbf{X}_1 \in \mathcal{X}_1} \mathcal{L}(\mathbf{X}_1, \mathbf{X}_2) \tag{6}$$

By Sion's minimax theorem [54], the set of Nash equilibria is nonempty. Then, given a Nash equilibrium $\mathbf{X}^*$, we define the *duality gap* of $\mathbf{X} = (\mathbf{X}_1, \mathbf{X}_2)$ as

$$\mathrm{Gap}_{\mathcal{L}}(\mathbf{X}) \coloneqq \mathcal{L}(\mathbf{X}_1^*, \mathbf{X}_2) - \mathcal{L}(\mathbf{X}_1, \mathbf{X}_2^*) \tag{7}$$

so $\mathrm{Gap}_{\mathcal{L}}(\mathbf{X}) \geq 0$ with equality if and only if $\mathbf{X}$ is itself a Nash equilibrium. In particular, $\mathbf{X}$ is an $\varepsilon$-Nash equilibrium of $\mathcal{Q}$ if and only if $\mathrm{Gap}_{\mathcal{L}}(\mathbf{X}) \leq \varepsilon$.

**Other semidefinite games.** In addition to quantum games, our framework can also be used for learning in other classes of games with a semidefinite structure as per [26, 45]. As an example, consider the problem of covariance matrix optimization in vector Gaussian multiple-access channels [9, 43, 57, 62]. In this case, there is a finite set of players indexed by $i \in \mathcal{N} = \{1, \ldots, N\}$; each player $i \in \mathcal{N}$ picks a unit-trace semidefinite matrix $\mathbf{X}_i \in \mathcal{X}_i$ and their payoff is given by the Shannon–Telatar capacity formula [57], viz.

$$u_i(\mathbf{X}_1, \ldots, \mathbf{X}_N) = \log \det\left(\mathbf{I} + \sum_j \mathbf{H}_j \mathbf{X}_j \mathbf{H}_j^\dagger\right) \tag{8}$$

where each $\mathbf{H}_i$ is a player-specific gain matrix [58]. Even though $u_i$ is no longer multilinear in $\mathbf{X}$, the algorithms we derive later in the paper can be applied to this setting essentially verbatim.

## 3   The matrix multiplicative weights algorithm

Throughout the sequel, we will focus on equilibrium learning in quantum – and semidefinite – games. In the context of two-player, zero-sum quantum games, the state-of-the-art method is based on the so-called *matrix multiplicative weights* (MMW) algorithm [7, 27, 29, 59] which updates as

$$\mathbf{Y}_{i,t+1} = \mathbf{Y}_{i,t} + \gamma_t \mathbf{V}_i(\mathbf{X}_t) \qquad \mathbf{X}_{i,t} = \frac{\exp(\mathbf{Y}_{i,t})}{\mathrm{tr}\left[\exp(\mathbf{Y}_{i,t})\right]} \tag{MMW}$$

In the above, (*a*) $\mathbf{X}_t = (\mathbf{X}_{1,t}, \mathbf{X}_{2,t})$ denotes the players' density matrix profile at each stage $t = 1, 2, \ldots$ of the process; (*b*) $\mathbf{V}_i(\mathbf{X}_t)$ is the payoff gradient of player $i \in \mathcal{N}$ under $\mathbf{X}_t$; (*c*) $\mathbf{Y}_t$ is an auxiliary state matrix that aggregates gradient steps over time; and (*d*) $\gamma_t > 0, t = 1, 2, \ldots$, is a learning rate (or step-size) parameter that can be freely tuned by the players.

Importantly, as stated, (MMW) requires *full information* at the player end: specifically, at each stage $t = 1, 2, \ldots$ of the process, each player $i \in \mathcal{N}$ must receive their individual payoff gradient $\mathbf{V}_i(\mathbf{X}_t)$ in order to perform the gradient update step in (MMW). Under this assumption, Jain & Watrous [27] showed that the induced empirical frequency of play

$$\bar{\mathbf{X}}_T = \frac{1}{T} \sum_{t=1}^{T} \mathbf{X}_t \tag{9}$$

converges to equilibrium at a rate of $\mathcal{O}(1/\sqrt{T})$ as per the formal result below:

**Theorem 1** (Jain & Watrous [27])**.** *Suppose that each player of a 2-player zero-sum game $\mathcal{Q}$ follows* (MMW) *for $T$ epochs with learning rate $\gamma = G^{-1}\sqrt{2H/T}$ where $H = \log(d_1 d_2)$. Then the players' empirical frequency of play enjoys the bound*

$$\text{Gap}_{\mathcal{L}}(\bar{\mathbf{X}}_T) \le G\sqrt{2H/T} \tag{10}$$

*In particular, if* (MMW) *is run for $T = \mathcal{O}(1/\varepsilon^2)$ iterations, $\bar{\mathbf{X}}_T$ will be an $\varepsilon$-Nash equilibrium of $\mathcal{Q}$.*

To the best of our knowledge, this guarantee of Jain & Watrous [27] remains the tightest known bound for Nash equilibrium learning in 2-player zero-sum quantum games. At the same time, Theorem 1 hinges on the players having perfect access to their individual gradients – which, among others, might entail full knowledge of the game, observing the other player's density matrix, etc. Our goal in the sequel will be to relax precisely this assumption and develop a payoff-based variant of (MMW) that can be employed without stringent information and observability requirements as above.

## 4   Matrix learning without matrix feedback

In an online learning framework, it is more realistic to assume that players observe only the *outcome* of their actions – i.e., their individual payoffs. In this information-starved, payoff-based setting, our main goal will be to employ a *minimal-information matrix multiplicative weights* (3MW) algorithm that updates as

$$\mathbf{Y}_{i,t+1} = \mathbf{Y}_{i,t} + \gamma_t \hat{\mathbf{V}}_{i,t} \qquad \mathbf{X}_{i,t} = \frac{\exp(\mathbf{Y}_{i,t})}{\text{tr}\big[\exp(\mathbf{Y}_{i,t})\big]} \tag{3MW}$$

where $\hat{\mathbf{V}}_{i,t}$ is some payoff-based estimate of the payoff gradient $\mathbf{V}_i(\mathbf{X}_t)$ of player $i$ at $\mathbf{X}_t$, and all other quantities are defined as per (MMW). In this regard, the main challenge that arises is how to reconstruct each player's payoff gradient matrices when they are not accessible via an oracle.

**4.1.   The classical approach: Importance weighted estimators.**   In the context of classical, finite games and multi-armed bandits, a standard approach for reconstructing $\hat{\mathbf{V}}_{i,t}$ is via the so-called *importance weighted estimator* (IWE) [12, 14, 32]. To state it in the context of finite games, assume that each player has at their disposal a finite set of *pure strategies* $\alpha_i \in \mathcal{A}_i$, and if each player plays $\hat{\alpha}_i \in \mathcal{A}_i$, then, in obvious notation, their individual payoff will be $\hat{u}_i = u_i(\hat{\alpha}_i; \hat{\alpha}_{-i})$. Then, if each player is using a mixed strategy $x_i \in \Delta(\mathcal{A}_i)$ to draw their chosen action $\hat{\alpha}_i$, the *importance weighted estimator* (IWE) for the payoff of the (possibly unplayed) action $\alpha_i \in \mathcal{A}_i$ of player $i$ is defined as

$$\text{IWE}_{i\alpha_i} = \frac{\mathbb{1}\{\alpha_i = \hat{\alpha}_i\}}{x_{i\alpha_i}} u_i(\hat{\alpha}_i; \hat{\alpha}_{-i}) \quad \text{for all } \alpha_i \in \mathcal{A}_i \tag{IWE}$$

with the assumption that $x_i$ has full support, i.e., each action $\alpha_i \in \mathcal{A}_i$ has strictly positive probability $x_{i\alpha_i}$ of being chosen by the $i$-th player.[1]

This approach has proven extremely fruitful in the context of multi-armed bandits and finite games where (IWE) is an essential ingredient of the optimal algorithms for each context [5, 12, 32, 65]. However, in our case, there are two insurmountable difficulties in extending (IWE) to a quantum context: First and foremost, the quantum regime is characterized by a *continuum* of pure states with highly correlated payoffs (in the sense that quantum states that are close in the Bloch sphere will have highly correlated POVM payoff observables); this comes in stark contrast to the classical regime of finite normal-form games, where players only have to contend with a finite number of actions (with no prior payoff correlations between them). Secondly, even after the realization of the pure states of the players, there is an inherent uncertainty and randomness due to the quantum measurement process that is invovled in the payoff-generating process; as such, the players' payoffs are also affected by an exogenous source of randomness which is altogether absent from (IWE).

Our approach to tackle these issues will be to exploit the reformulation of a quantum game as a continuous game with multilinear payoffs over the spectraplex (or, rather, a product thereof), and use ideas from bandit convex optimization – in the spirit of [21, 31] – to estimate the players' payoff gradients with minimal, scalar information requirements.

---

[1]The assumption that $x_{i,t}$ has full support is only for technical reasons. In practice, it can be relaxed by using IWE with explicit exploration – see [32] for more details.

## 4.2. Gradient estimation via finite-difference quotients on the spectraplex.

To provide some intuition for the analysis to come, consider first a single-variable smooth function $f : \mathbb{R} \to \mathbb{R}$ and a point $x \in \mathbb{R}$. Then, for error tolerance $\delta > 0$, a two-point estimate of the derivative of $f$ at $x$ is given by the expression

$$\hat{f}_x = \frac{f(x + \delta) - f(x - \delta)}{2\delta} \tag{11}$$

Going to higher dimensions, letting $f : \mathbb{R}^d \to \mathbb{R}$ be a smooth function, $\{e_1, \dots, e_d\}$ be the standard basis of $\mathbb{R}^d$ and $s$ drawn from $\{e_1, \dots, e_d\}$ uniformly at random, the estimator

$$\hat{f}_x = \frac{d}{2\delta} [f(x + \delta s) - f(x - \delta s)] s \tag{12}$$

is a $\mathcal{O}(\delta)$-approximation of the gradient, i.e., $\|\mathbb{E}_s[\hat{f}_x] - \nabla f(x)\|_F = \mathcal{O}(\delta)$. This idea is the basis of the Kiefer–Wolfowitzs stochastic approximation scheme [30] and will be the backbone of our work.

Now, to employ this type of estimator for a function over the set of density matrices $\mathcal{X}$ in $\mathbb{H}^d$, we need to ensure two things: *(i)* the feasibility of the *sampling direction*, and *(ii)* the feasibility of the *evaluation point*. The first caveat is due to the fact that the set of the density matrices forms a lower dimensional manifold in the set of Hermitian operators, and therefore, not all directions from a base of $\mathbb{H}^d$ are feasible. The second one is due to the fact that $\mathcal{X}$ is bounded, thus, even if the sampling direction is feasible, the evaluation point can lie outside the set $\mathcal{X}$. We proceed to ensure all this in a series of concrete steps below.

**Sampling Directions.** We begin with the issue of defining a proper sampling set for the estimator's finite-difference directions. To that end, we will first construct an orthonormal basis of the tangent hull $\mathcal{Z} = \{\mathbf{Z} \in \mathbb{H}^d : \operatorname{tr} \mathbf{Z} = 0\}$ of $\mathcal{X}$, i.e., the subspace of traceless matrices of $\mathbb{H}^d$. Note that if $\mathbf{Z} \in \mathcal{Z}$ then for any $\mathbf{X} \in \mathbb{H}^d$ it holds *(a)* $\mathbf{X} + \mathbf{Z} \in \mathbb{H}^d$, and *(b)* $\operatorname{tr}[\mathbf{X} + \mathbf{Z}] = \operatorname{tr}[\mathbf{X}]$.

Denoting by $\Delta_{k\ell} \in \mathbb{H}^d$ the matrix with 1 in the $(k, \ell)$-position and 0's everywhere else, it is easy to see that the set $\left\{ \{\Delta_{jj}\}_{j=1}^d, \{\mathbf{e}_{k\ell}\}_{k<\ell}, \{\tilde{\mathbf{e}}_{k\ell}\}_{k<\ell} \right\}$ is an orthonormal basis of $\mathbb{H}^d$, where

$$\mathbf{e}_{k\ell} = \frac{1}{\sqrt{2}} \Delta_{k\ell} + \frac{1}{\sqrt{2}} \Delta_{\ell k} \quad \text{and} \quad \tilde{\mathbf{e}}_{k\ell} = \frac{i}{\sqrt{2}} \Delta_{k\ell} - \frac{i}{\sqrt{2}} \Delta_{\ell k} \tag{13}$$

for $1 \le k < \ell \le d$, where $i$ is the imaginary unit with $i^2 = -1$. The next proposition provides a basis for the subspace $\mathcal{Z}$, whose proof lies in the appendix.

**Proposition 1.** *Let $\mathbf{E}_j$ be defined as* $\mathbf{E}_j = \frac{1}{\sqrt{j(j+1)}} \left( \Delta_{11} + \cdots + \Delta_{jj} - j\Delta_{j+1,j+1} \right)$ *for* $j = 1, \dots, d-1$. *Then, the set* $\mathcal{E} = \left\{ \{\mathbf{E}_j\}_{j=1}^{d-1}, \{\mathbf{e}_{k\ell}\}_{k<\ell}, \{\tilde{\mathbf{e}}_{k\ell}\}_{k<\ell} \right\}$ *is an orthonormal basis of $\mathcal{Z}$.*

In the sequel, we will use this basis as an orthnormal sampler from which to pick the finite-difference directions for the estimation of $\mathbf{V}$.

**Feasibility Adjustment.** After establishing an orthonormal basis for $\mathcal{Z}$ as per Proposition 1, we readily get that for any $\mathbf{X} \in \mathcal{X}$, any $\mathbf{Z} \in \mathcal{E}^{\pm} := \left\{ \{\pm\mathbf{E}_j\}_{j=1}^{d-1}, \{\pm\mathbf{e}_{k\ell}\}_{k<\ell}, \{\pm\tilde{\mathbf{e}}_{k\ell}\}_{k<\ell} \right\}$ and $\delta > 0$, the point $\mathbf{X} + \delta\mathbf{Z}$ belongs to $\mathcal{Z}$. However, depending on the value of the exploration parameter $\delta$ and the distance of $\mathbf{X}$ from the boundary of $\mathcal{X}$, the point $\mathbf{X} + \delta\mathbf{Z} \in \mathbb{H}^d$ may fail to lie in $\mathcal{X}$ due to violation of the positive-semidefinite condition. On that account, we now treat the latter restriction, i.e., the feasibility of the *evaluation point*.

To tackle this, the idea is to transfer the point $\mathbf{X}$ toward the interior of $\mathcal{X}$ and move along the sampled direction from there. For this, we need to find a reference point $\mathbf{R} \in \operatorname{ri}(\mathcal{X})$ and a "safety net" $r > 0$ such that $\mathbf{R} + r\mathbf{Z} \in \mathcal{X}$ for any $\mathbf{Z} \in \mathcal{E}^{\pm}$. Then, for $\delta \in (0, r)$, the point

$$\mathbf{X}^{(\delta)} := \mathbf{X} + \frac{\delta}{r}(\mathbf{R} - \mathbf{X}) \tag{14}$$

lies in $\operatorname{ri}(\mathcal{X})$, and moving along $\mathbf{Z} \in \mathcal{E}^{\pm}$, the point $\mathbf{X}^{(\delta)} + \delta\mathbf{Z} = (1 - \frac{\delta}{r})\mathbf{X} + \frac{\delta}{r}(\mathbf{R} + r\mathbf{Z})$ remains in $\mathcal{X}$ as a convex combination of two elements in $\mathcal{X}$. The following proposition provides an exact expression for $\mathbf{R}$ and $r$, which we will use next to guarantee the feasibility of the sampled iterates.

**Proposition 2.** *Let $\mathbf{R} = \frac{1}{d} \sum_{j=1}^d \Delta_{jj}$. Then, for $r = \min\left\{ \frac{1}{\sqrt{d(d-1)}}, \frac{\sqrt{2}}{d} \right\}$, it holds that $\mathbf{R} + r\mathbf{Z} \in \mathcal{X}$ for any direction $\mathbf{Z} \in \mathcal{E}^{\pm}$.*

# 5 Bandit learning in zero-sum quantum games

With all these in hand, we are now ready to proceed to the presentation of the MMW with limited feedback information. To streamline our presentation, before delving into the more difficult "bandit feedback" case – where each player $i \in \mathcal{N}$ only observes the realized payoff observable $U_i(\omega)$ – we begin with the simpler case where players observe their mixed payoffs $u_i$ at a given profile $\mathbf{X} \in \mathcal{X}$.

**5.1. Learning with mixed payoff observations.** Our main idea to exploit the observation of mixed payoffs and the finite-difference sampling to the fullest will be to introduce a "coordination phase" where players take a sampling step before updating their state variables and continue playing. In more detail, we will take an approach similar to Bervoets et al. [8] and assume that players alternate between an "exploration" and an "exploitation" update that allows them to sample the landscape of $\mathcal{L}$ efficiently at each iteration. Concretely, writing $\mathbf{X}_t$ and $\delta_t$ for the players' state profile and sampling radius $\delta_t$ at stage $t = 1, 2, \ldots$, the sequence of events that we envision proceeds as follows:

**Step 1.** Draw a sampling direction $\mathbf{Z}_{i,t} \in \mathcal{E}_i$ and $s_{i,t} \in \{\pm 1\}$ uniformly at random.

**Step 2.** (a) Play $\mathbf{X}_{i,t}^{(\delta)} + s_{i,t}\,\delta_t\,\mathbf{Z}_{i,t}$ and observe $u_i(\mathbf{X}_t^{(\delta)} + s_t\delta_t\mathbf{Z}_t)$.

     (b) Play $\mathbf{X}_{i,t}^{(\delta)} - s_{i,t}\,\delta_t\,\mathbf{Z}_{i,t}$ and observe $u_i(\mathbf{X}_t^{(\delta)} - s_t\delta_t\mathbf{Z}_t)$.

**Step 3.** Approximate $\mathbf{V}_i(\mathbf{X}_t)$ via the *two-point estimator* (2PE):

$$\hat{\mathbf{V}}_{i,t} \coloneqq \frac{D_i}{2\delta_t}\left[u_i(\mathbf{X}_t^{(\delta)} + s_t\delta_t\mathbf{Z}_t) - u_i(\mathbf{X}_t^{(\delta)} - s_t\delta_t\mathbf{Z}_t)\right] s_{i,t}\mathbf{Z}_{i,t} \tag{2PE}$$

where $D_i = d_i^2 - 1$ is the dimension of $\mathbb{H}^{d_i}$, and $D \coloneqq \max_{i \in \mathcal{N}} D_i$.

The main guarantee of the resulting (3MW) + (2PE) algorithm may then be stated as follows:

**Theorem 2.** *Suppose that each player of a 2-player zero-sum game $\mathcal{Q}$ follows* (3MW) *for $T$ epochs with learning rate $\gamma$, sampling radius $\delta$, and gradient estimates provided by* (2PE)*. Then the players' empirical frequency of play enjoys the duality gap guarantee*

$$\mathbb{E}\left[\mathrm{Gap}_{\mathcal{L}}(\bar{\mathbf{X}}_T)\right] \leq \frac{H}{\gamma T} + 8D^2G^2\gamma + 16DL\delta \tag{15}$$

*where $H = \log(d_1 d_2)$. In particular, for $\gamma = (DG)^{-1}\sqrt{H/(8T)}$ and $\delta = (G/L)\sqrt{H/(8T)}$, the players enjoy the equilibrium convergence guarantee*

$$\mathbb{E}\left[\mathrm{Gap}_{\mathcal{L}}(\bar{\mathbf{X}}_T)\right] \leq 8DG\sqrt{2H/T}. \tag{16}$$

Compared to Theorem 1, the convergence rate (16) of Theorem 2 is quite significant because it only differs by a factor which is linear in the dimension of the ambient space and otherwise maintains the same $\mathcal{O}(\sqrt{T})$ dependence on the algorithm's runtime. In this regard, Theorem 2 shows that the "explore-exploit" sampler underlying (2PE) is essentially as powerful as the full information framework of Jain & Watrous [27] – and this, despite the fact that players no longer require access to the gradient matrix $\mathbf{V}$ of $\mathcal{L}$. This echoes a range of previous findings in stochastic convex optimization for the efficiency of two-point samplers [3, 53], a similarity we find particularly surprising given the stark differences between the two settings – non-commutativity, min-max versus min-min landscape. The key ingredients for the equilibrium convergence rate of Theorem 2 are the two technical results below. The first is a feedback-agnostic "energy inequality" which is tied to the update structure of (MMW) and is stated in terms of the quantum relative entropy function

$$D(\mathbf{P}, \mathbf{X}) = \mathrm{tr}[\mathbf{P}(\log \mathbf{P} - \log \mathbf{X})] \tag{17}$$

for $\mathbf{P}, \mathbf{X} \in \mathcal{X}$ with $\mathbf{X} \succ 0$. Concretely, we have the following estimate.

**Lemma 1.** *Fix some $\mathbf{P} \in \mathcal{X}$, and let $\mathbf{X}_t, \mathbf{X}_{t+1}$ be two successive iterates of* (3MW)*, without any assumptions for the input sequence $\hat{\mathbf{V}}_t$. We then have*

$$D(\mathbf{P}, \mathbf{X}_{t+1}) \leq D(\mathbf{P}, \mathbf{X}_t) + \gamma_t\,\mathrm{tr}[\hat{\mathbf{V}}_t(\mathbf{X}_t - \mathbf{P})] + \frac{\gamma_t^2}{2}\|\hat{\mathbf{V}}_t\|_F^2. \tag{18}$$

The proof of Lemma 1 follows established techniques in the theory of (MMW), so we defer a detailed discussion to the appendix. The second result that we will need is tailored to the estimator (2PE) and provides a tight estimate of its moments conditioned on the history $\mathcal{F}_t = \mathcal{F}(\mathbf{X}_1, \ldots, \mathbf{X}_t)$ of $\mathbf{X}_t$.

---

**Algorithm 1:** MMW with bandit feedback

---
1: **Input:** $\mathbf{Y}_1 \leftarrow 0$; safety parameter $r_i$ and anchor point $\mathbf{R}_i$, $i \in \mathcal{N}$; step-size $\gamma_t$; sampling radius $\delta_t$
2: **for** $t = 1, 2, \ldots$ **do simultaneously** for all $i \in \mathcal{N}$
3:     Set $\mathbf{X}_{i,t} = \exp(\mathbf{Y}_{i,t})/\operatorname{tr}[\exp(\mathbf{Y}_{i,t})]$.
4:     **Sample** $\mathbf{Z}_{i,t}$ uniformly from $\mathcal{E}_i^\pm$.
5:     **Play** $\mathbf{X}_{i,t}^{(\delta)} + \delta_t \mathbf{Z}_{i,t}$.
6:     **Observe** $U_i(\omega_t)$.
7:     Set $\hat{\mathbf{V}}_{i,t} \coloneqq D_i/\delta_t \cdot U_i(\omega_t)\mathbf{Z}_{i,t}$.
8:     **Update** $\mathbf{Y}_{i,t+1} \leftarrow \mathbf{Y}_{i,t} + \gamma_t \hat{\mathbf{V}}_{i,t}$.
9: **end for**

---

**Proposition 3.** *The estimator* (2PE) *enjoys the conditional bounds*

$$(i) \quad \left\| \mathbb{E}[\hat{\mathbf{V}}_t \mid \mathcal{F}_t] - \mathbf{V}(\mathbf{X}_t) \right\|_F \le 4DL\delta_t \quad and \quad (ii) \quad \mathbb{E}\big[\|\hat{\mathbf{V}}_t\|_F^2 \,\big|\, \mathcal{F}_t\big] \le 16D^2G^2 \tag{19}$$

The defining element in Proposition 3 is that even though the estimator (2PE) is biased, its second moment is bounded as $\mathcal{O}(1)$. This is ultimately due to the multilinearity of the players' payoff functions and plays a pivotal role in showing that the duality gap of $\bar{\mathbf{X}}_t$ under (3MW) is of the same order as under (MMW), because the bias can be controlled with affecting the variance of the estimator. We provide a detailed proof of Lemma 1, Proposition 3, and Theorem 2 in the appendix.

**5.2. Learning with bandit feedback.** Despite its strong convergence guarantees, a major limiting factor in the applicability of Theorem 2 is that, in many cases, the game's players may only be able to observe their realized payoff observables $U_i(\omega)$, and their mixed payoffs $u_i(\mathbf{X})$ could be completely inaccessible. In particular, as we described in Section 2, each outcome $\omega \in \Omega$ of the POVM occurs with probability $P_\omega(\mathbf{X}_t)$ under the strategy profile $\mathbf{X}_t$. Accordingly, if this is the only information available to the players, they will need to estimate their individual payoff gradients through the single observation of the (random) scalar $U_i(\omega_t) \in \mathbb{R}$. In view of this, and inspired by previous works on payoff-based learning and zeroth-order optimization [8, 9, 11, 25, 49, 50], we will consider the single-point stochastic approximation approach of [21, 55] which unfolds as follows:

**Step 1.** Each player draws a sampling direction $\mathbf{Z}_{i,t} \in \mathcal{E}_i^\pm$ uniformly at random.

**Step 2.** Each player plays $\mathbf{X}_{i,t}^{(\delta)} + \delta_t \mathbf{Z}_{i,t}$.

**Step 3.** Each player receives $U_i(\omega_t)$.

**Step 4.** Each player approximates $\mathbf{V}_i(\mathbf{X}_t)$ via the the *one-point estimator* (1PE):

$$\hat{\mathbf{V}}_{i,t} \coloneqq \frac{D_i}{\delta_t} U_i(\omega_t)\, \mathbf{Z}_{i,t} \tag{1PE}$$

In this case, the players' gradient estimates may be bounded as follows:

**Proposition 4.** *The estimator* (1PE) *enjoys the conditional bounds*

$$(i) \quad \|\mathbb{E}[\hat{\mathbf{V}}_t \mid \mathcal{F}_t] - \mathbf{V}(\mathbf{X}_t)\|_F \le 4DL\delta_t \quad and \quad (ii) \quad \mathbb{E}[\|\hat{\mathbf{V}}_t\|_F^2 \mid \mathcal{F}_t] \le 4D^2B^2/\delta_t^2. \tag{20}$$

The crucial difference between Propositions 3 and 4 is that the former leads to a gradient estimator with $\mathcal{O}(1)$ variance and magnitude, whereas the magnitude of the latter is inversely proportional to $\delta_t$; however, since $\delta_t$ in turn controls the *bias* of the gradient estimator, we must now resolve a bias-variance dilemma, which was absent in the case of (2PE). This leads to the following variant of Theorem 2 with bandit, realization-based feedback:

**Theorem 3.** *Suppose that each player of a 2-player zero-sum game $\mathcal{Q}$ follows* (3MW) *for $T$ epochs with learning rate $\gamma$, sampling radius $\delta$, and gradient estimates provided by* (1PE). *Then the players' empirical frequency of play enjoys the duality gap guarantee*

$$\mathbb{E}\big[\operatorname{Gap}_{\mathcal{L}}(\bar{\mathbf{X}}_T)\big] \le \frac{H}{\gamma T} + \frac{2D^2B^2\gamma}{\delta^2} + 16DL\delta \tag{21}$$

*where $H = \log(d_1 d_2)$. In particular, for $\gamma = \left(\frac{H}{2T}\right)^{3/4} \frac{1}{2D\sqrt{BL}}$ and $\delta = \left(\frac{H}{2T}\right)^{1/4} \sqrt{\frac{B}{4L}}$, the players enjoy the equilibrium convergence guarantee:*

$$\mathbb{E}\big[\operatorname{Gap}_{\mathcal{L}}(\bar{\mathbf{X}}_T)\big] \le \frac{2^{3/4}\, 8H^{1/4}D\sqrt{BL}}{T^{1/4}}. \tag{22}$$

An important observation here is that the players' equilibrium convergence rate under (3MW)+(1PE) no longer matches the convergence rate of the vanilla MMW algorithm (Theorem 1). The reason for this is the bias-variance trade-off in the estimator (1PE), and is reminiscent of the drop in the rate of regret minimization from $\mathcal{O}(T^{1/2})$ to $\mathcal{O}(T^{2/3})$ under (IWE) with bandit feedback and explicit exploration in finite games. A kernel-based approach in the spirit of Bubeck et al. [13] could possibly be used to fill the $\mathcal{O}(T^{1/4})$ gap between Theorems 1 and 3, but this would come at the cost of a possibly catastrophic dependence on the dimension (which is already quadratic in our setting). This consideration is beyond the scope of our work, but it would constitute an important future direction.

## 6    Bandit learning in $N$-player quantum games

We conclude our paper with an examination of the behavior of the MMW algorithm in general, $N$-player quantum games. Here, a major difficulty that arises is that, in stark contrast to the min-max case, the set of the game's equilibria can be disconnected, so any convergence result will have to be, by necessity, local. In addition, because general $N$-games do not have the amenable profile of a bilinear min-max problem – they are multilinear, multi-objective problems – it will not be possible to obtain any convergence guarantees for the game's empirical frequency of play (since there is no convex structure to exploit). Instead, we will have to focus squarely on the induced trajectory of play, which carries with it a fair share of complications.

Inspired by the very recent work of [38], we will not constrain our focus to a specific class of *games*, but to a specific class of *equilibria*. In particular, we will consider the behavior of MMW-based learning with respect to Nash equilibria $\mathbf{X}^* \in \mathcal{X}$ that satisfy the *variational stability* condition

$$\mathrm{tr}[\mathbf{V}(\mathbf{X})(\mathbf{X} - \mathbf{X}^*)] < 0 \quad \text{for all } \mathbf{X} \in \mathcal{U}\backslash\{\mathbf{X}^*\}. \tag{VS}$$

This condition can be traced back to [44], and can be seen as a game-theoretic analogue of first-order stationarity in the context of continuous optimization, or as an equilibrium refinement in the spirit of the seminal concept of *evolutionary stability* in population games [39, 40].[2] Importantly, as was shown in [38], variationally stable equilibria are the only equilibria that are asymptotically stable under the *continuous-time* dynamics of the "follow the regularized leader" (FTRL) class of learning policies, so it stands to reason to ask whether they enjoy a similar convergence landscape in the context of bona fide, discrete-time learning with minimal, payoff-based feedback.

Our final result provides an unambiguously positive answer to this question:[3]

**Theorem 4.** *Fix some tolerance level $\eta \in (0, 1)$ and suppose that the players of an $N$-player quantum game follow* (3MW) *with bandit, realization-based feedback, and surrogate gradients provided by the estimator* (1PE) *with step-size and sampling radius parameters such that*

$$(i)\ \textstyle\sum_{t=1}^{\infty} \gamma_t = \infty, \quad (ii)\ \textstyle\sum_{t=1}^{\infty} \gamma_t \delta_t < \infty, \quad and \quad (ii)\ \textstyle\sum_{t=1}^{\infty} \gamma_t^2/\delta_t^2 < \infty. \tag{23}$$

*If $\mathbf{X}^*$ is variationally stable, there exists a neighborhoold $\mathcal{U}$ of $\mathbf{X}^*$ such that*

$$\mathbb{P}(\lim_{t\to\infty} \mathbf{X}_t = \mathbf{X}^*) \geq 1 - \eta \quad whenever\ \mathbf{X}_1 \in \mathcal{U}. \tag{24}$$

It is worth noting that the last-iterate convergence guarantee of Theorem 4 is considerably stronger than the time-averaged variants of Theorems 1–3, and we are not aware of any comparable convergence guarantee for general quantum games. [Trivially, last-iterate convergence implies time-averaged convergence, but the converse, of course, may fail to hold] As such, especially in cases that require to track the trajectory of the system or the players' day-to-day rewards, Theorem 4 provides an important guarantee for the realized sequence of events.

On the other hand, in contrast to Theorem 4, it should be noted that the guarantees of Theorems 1–3 are global. Given that general quantum games may in general possess a large number of disjoint Nash equilibria, this transition from global to local convergence guarantees seems unavoidable. It is, however, an open question whether (VS) could be exploited further in order to deduce the rate of convergence to such equilibria; we leave this as a direction for future research.

---

[2]It should be noted here that, if reduced to the simplex, the stability condition (VS) is exactly equivalently to the variational characterization of evolutionarily stable states due to Taylor [56].

[3]Strictly speaking, the algorithms (3MW) and (1PE) have been stated in the context of 2-player games. The extension to $N$-player games is straightforward, so we do not present it here; for the details (which hide no subtleties), see the appendix.

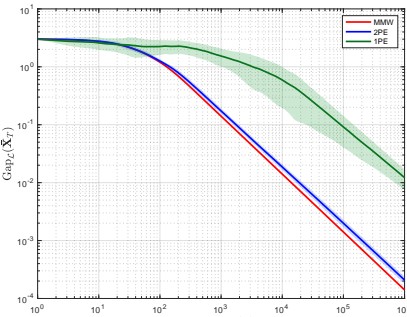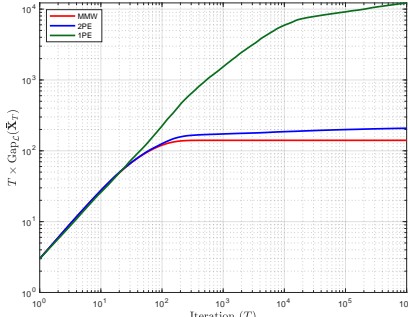

**Figure 1:** Performance evaluation of the (3MW) with the (2PE) and (1PE) estimators and comparison with the full information (MMW). The solid lines correspond to the mean values of the duality gap of each method, and the shaded regions enclose the area of ±1 (sample) standard deviation among the different runs.

## 7 Numerical Experiments

In this last section, we provide numerical simulations to validate and explore the performance of (MMW) with payoff-based feedback. Additional experiments can be found in Appendix E.

**Game setup.** Our testbed is a two-player zero-sum quantum game, which is the quantum analogue of a $2 \times 2$ min-max game with actions $\{\alpha_1, \alpha_2\}$ and $\{\beta_1, \beta_2\}$, and payoff matrix

$$P = \begin{pmatrix} (4, -4) & (2, -2) \\ (-4, 4) & (-2, 2) \end{pmatrix} \tag{25}$$

In the quantum regime, the payoff information of the quantum game is encoded in the Hermitian matrices $\mathbf{W}_1 = \mathrm{diag}(4, 2, -4, -2)$, and $\mathbf{W}_2 = -\mathbf{W}_1$ as per Eq. (3) in Section 2. By elementary considerations, the action profile $(\alpha_1, \beta_2)$ is a strict Nash equilibrium of the classical zero-sum game, which corresponds to the pure quantum state with density matrix profile $\mathbf{X}^* = (\mathbf{X}_1^*, \mathbf{X}_2^*)$ where $\mathbf{X}_1^* = e_1 \otimes e_1$ and $\mathbf{X}_2^* = e_2 \otimes e_2$ in the standard basis in which $\mathbf{W}_1$ and $\mathbf{W}_2$ are diagonal.

**Convergence speed analysis.** In Fig. 1, we evaluate the convergence properties of (3MW) using the estimators (2PE) and (1PE), and compare it with the full information variant (MMW). For each method, we perform 10 different runs, with $T = 10^5$ steps each, and compute the mean value of the duality gap as a function of the iteration $t = 1, 2, \ldots, T$. The solid lines correspond to the mean values of the duality gap of each method, and the shaded regions enclose the area of ±1 (sample) standard deviation among the 10 different runs. Note that the red line, which corresponds to the full information (MMW), does not have a shaded region, since there is no randomness in the algorithm. All the runs for the three different methods were initialized for $\mathbf{Y} = 0$ and we used $\gamma = 10^{-2}$ for all methods. In particular, for (3MW) with gradient estimates given by (2PE) estimator, we used a sampling radius $\delta = 10^{-2}$, and for (3MW) with (1PE) estimator, we used $\delta = 10^{-1}$ (in tune with our theoretical results which suggest the use of a tighter sampling radius when mixed payoff information is available to the players).

Figure 1 has several important take-aways. First and foremost, as is to be expected, the payoff-based methods lag behind the full-information variant of (MMW); however, what is particularly surprising is that the drop in performance is singularly mild. As we see in the second plot in Fig. 1, the various algorithms achieved a rate of convergence closer to $\mathcal{O}(1/T)$, which is significantly faster than $\mathcal{O}(1/\sqrt{T})$ and/or $\mathcal{O}(1/T^{1/4})$. This suggests that, in practice, the bandit variants of (MMW) may yield excellent performance benefits, despite the high degree of uncertainty incurred by the complete lack of information on the game being played.

## Acknowledgments and Disclosure of Funding

This work has been partially supported by the Air Force Office of Scientific Research under award number FA9550-20-1-0397. Additional support is gratefully acknowledged from NSF 1915967, 2118199, 2229012, 2312204. KL is grateful for financial support by the Onassis Foundation (F ZR 033-1/2021-2022). PM is also with the Archimedes Research Unit – Athena RC – University of Athens, and gratefully acknowledges financial support by project MIS 5154714 of the National Recovery and Resilience Plan Greece 2.0 funded by the European Union under the NextGenerationEU Program, and by the French National Research Agency (ANR) in the framework of the "Investissements d'avenir" program (ANR-15-IDEX-02), the LabEx PERSYVAL (ANR-11-LABX-0025-01), MIAI@Grenoble Alpes (ANR-19-P3IA-0003). NB was supported by the Koret Foundation via the Digital Living 2030 project.

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

# Appendix

In the series of technical appendices that follow, we provide the missing proofs from the main part of our paper, and we provide some numerical illustrations of the performance of the proposed algorithms. As a roadmap, we begin in Appendix A with some auxiliary results that are required throughout our analysis. Subsequently, in Appendices B–D, we provide the proofs of the results presented in Sections 4–6 respectively. Finally, in Appendix E, we provide a suite of numerical experiments to assess the practical performance of (3MW) using the estimators (2PE) and (1PE), and we compare it with the full information setting underlying (MMW).

## A   Auxiliary Results

We now introduce some notation for quantum games in a $N$-player setting, and explain how the extension from the 2-player setting is straightforward.

**$N$-player quantum games.**   First of all, a quantum game $\mathcal{Q}$ consists of a finite set of players $i \in \mathcal{N} = \{1, \ldots, N\}$, where each player $i \in \mathcal{N}$ has access to a complex Hilbert space $\mathcal{H}_i \cong \mathbb{C}^{d_i}$. The set of pure states is the unit sphere $\Psi_i \coloneqq \{\psi_i \in \mathcal{H}_i : \|\psi_i\|_F = 1\}$ of $\mathcal{H}_i$. We will write $\Psi \coloneqq \prod_i \Psi_i$ for the space of all ensembles $\psi = (\psi_1, \ldots, \psi_N)$ of pure states $\psi_i \in \Psi_i$ that are independently prepared by each $i \in \mathcal{N}$.

In analogy with the 2-player case, each outcome $\omega \in \Omega$ is associated to a positive semi-definite operator $\mathbf{P}_\omega \colon \mathcal{H} \to \mathcal{H}$ defined on the tensor product $\mathcal{H} \coloneqq \otimes_i \mathcal{H}_i$ of the players' individual state spaces; we further assume that $\sum_{\omega \in \Omega} \mathbf{P}_\omega = \mathbf{I}$, thus, the probability of observing $\omega \in \Omega$ at state $\psi \in \Psi$ is

$$P_\omega(\psi) = \langle \psi_1 \otimes \cdots \otimes \psi_N | \mathbf{P}_\omega | \psi_1 \otimes \cdots \otimes \psi_N \rangle \tag{A.1}$$

and, the player's expected payoff at state $\psi \in \Psi$ will be

$$u_i(\psi) \coloneqq \langle U_i \rangle \equiv \sum_\omega P_\omega(\psi)\, U_i(\omega) \tag{A.2}$$

Similarly to the 2-player setting, if each player $i \in \mathcal{N}$ prepares a density matrix $\mathbf{X}_i$ as per (1), the expected payoff of player $i \in \mathcal{N}$ under $\mathbf{X} = (\mathbf{X}_1, \ldots, \mathbf{X}_N)$ will be

$$u_i(\mathbf{X}) = \sum_{\omega \in \Omega} U_i(\omega)\, \mathrm{tr}[\mathbf{P}_\omega \mathbf{X}_1 \otimes \cdots \otimes \mathbf{X}_N] = \mathrm{tr}[\mathbf{W}_i \mathbf{X}_1 \otimes \cdots \otimes \mathbf{X}_N] \tag{A.3}$$

where $\mathbf{W}_i = \sum_{\omega \in \Omega} U_i(\omega) \mathbf{P}_\omega \in \mathcal{H}$ for $i \in \mathcal{N}$. Finally, we denote by $\mathbf{V}_i(\mathbf{X})$ the individual payoff gradient of player $i$ under $\mathbf{X}$ as

$$\mathbf{V}_i(\mathbf{X}) \coloneqq \nabla_{\mathbf{X}_i^\top} u_i(\mathbf{X}) \tag{A.4}$$

All other notions are extended, accordingly.                                                                                    ◊

As noted in Section 2, we define the norm $\|A\|_F = \sqrt{\mathrm{tr}[A^\dagger A]}$ for any $A \in \mathbb{H}^{d_i}$, i.e., $(\mathbb{H}^{d_i}, \|\cdot\|_F)$ is an inner-product space. With a slight abuse of notation, we define for $\mathbf{X} = (\mathbf{X}_1, \ldots, \mathbf{X}_N) \in \mathcal{X}$ its norm as:

$$\|\mathbf{X}\|_F = \sqrt{\sum_{i=1}^N \|\mathbf{X}_i\|_F^2} \tag{A.5}$$

**Lemma A.1.** *For any $\mathbf{X}_i \in \mathcal{X}_i$, it holds $\|\mathbf{X}_i\|_F \leq 1$, and $\mathrm{diam}(\mathcal{X}) = 2\sqrt{N}$.*

*Proof.* For the first part, since $\mathbf{X}_i \in \mathcal{X}_i$, it admits an orthonormal decomposition $Q\Lambda Q^\dagger$ such that $QQ^\dagger = Q^\dagger Q = \mathbf{I}$ and $\Lambda = \mathrm{diag}(\lambda_1, \ldots, \lambda_{d_i})$ with $\sum_{j=1}^{d_i} \lambda_j = 1$, and $\lambda_j \geq 0$ for all $j$. Hence

$$\|\mathbf{X}_i\|_F^2 = \mathrm{tr}[\mathbf{X}_i^\dagger \mathbf{X}_i] = \mathrm{tr}[Q\Lambda Q^\dagger Q\Lambda Q^\dagger] = \mathrm{tr}[Q\Lambda^2 Q^\dagger] = \sum_{j=1}^{d_i} \lambda_i^2 \leq \sum_{j=1}^{d_i} \lambda_i = 1 \tag{A.6}$$

where the last inequality holds, since $0 \leq \lambda_j \leq 1$, and the result follows.

For the second part, letting $\mathbf{X} = (\mathbf{X}_1, \ldots, \mathbf{X}_N)$ and $\mathbf{X}' = (\mathbf{X}'_1, \ldots, \mathbf{X}'_N)$ be two points in $\mathcal{X}$, we have

$$\|\mathbf{X} - \mathbf{X}'\|_F = \sqrt{\sum_{i=1}^{N} \|\mathbf{X}_i - \mathbf{X}'_i\|_F^2} \leq \sqrt{\sum_{i=1}^{N} (2\|\mathbf{X}_i\|_F^2 + 2\|\mathbf{X}'_i\|_F^2)} \leq 2\sqrt{N} \tag{A.7}$$

and since the equality is attained, we get the result. ∎

Our next result concerns the *quantum relative entropy*

$$D(\mathbf{P}, \mathbf{X}) = \sum_{i=1}^{N} D_i(\mathbf{P}_i, \mathbf{X}_i) \tag{A.8}$$

where $\mathbf{P} = (\mathbf{P}_1, \ldots, \mathbf{P}_N) \in \mathcal{X}$ and $\mathbf{X} = (\mathbf{X}_1, \ldots, \mathbf{X}_N) \in \mathrm{ri}(\mathcal{X})$ and

$$D_i(\mathbf{P}_i, \mathbf{X}_i) := \mathrm{tr}[\mathbf{P}_i(\log \mathbf{P}_i - \log \mathbf{X}_i)] \tag{A.9}$$

The lemma we will require is a semidefinite version of Pinsker's inequality which reads as follows:

**Lemma A.2.** *For all* $\mathbf{P} \in \mathcal{X}$ *and* $\mathbf{X} \in \mathrm{ri}(\mathcal{X})$ *we have*

$$D(\mathbf{P}, \mathbf{X}) \geq \frac{1}{2}\|\mathbf{P} - \mathbf{X}\|_F^2 \tag{A.10}$$

*Proof.* Focusing on player $i \in \mathcal{N}$, we will show first that

$$D_i(\mathbf{P}_i, \mathbf{X}_i) \geq \frac{1}{2}\|\mathbf{P}_i - \mathbf{X}_i\|_F^2 \tag{A.11}$$

for all $\mathbf{P} = (\mathbf{P}_1, \ldots, \mathbf{P}_N) \in \mathcal{X}$ and $\mathbf{X} = (\mathbf{X}_1, \ldots, \mathbf{X}_N) \in \mathrm{ri}(\mathcal{X})$.

To this end, we define the function $h_i \colon \mathbb{H}_+^{d_i} \to \mathbb{R}$ as $h_i(\mathbf{X}_i) = \mathrm{tr}[\mathbf{X}_i \log \mathbf{X}_i]$, which is 1-strongly convex with respect to the nuclear norm $\|\cdot\|_1$ [63], and since $\|\mathbf{X}_i\|_1 \geq \|\mathbf{X}_i\|_F$ for all $\mathbf{X}_i \in \mathcal{X}_i$, we readily get that $h_i$ is 1-strongly convex with respect to the Frobenius norm, as well.

Letting $\nabla h_i(\mathbf{X}_i) = \log \mathbf{X}_i + \mathbf{I}$, by 1-strong convexity, we have for $\mathbf{P} = (\mathbf{P}_1, \ldots, \mathbf{P}_N) \in \mathcal{X}$ and $\mathbf{X} = (\mathbf{X}_1, \ldots, \mathbf{X}_N) \in \mathrm{ri}(\mathcal{X})$:

$$h_i(\mathbf{P}_i) \geq h_i(\mathbf{X}_i) + \mathrm{tr}[\nabla h_i(\mathbf{X}_i)(\mathbf{P}_i - \mathbf{X}_i)] + \frac{1}{2}\|\mathbf{P}_i - \mathbf{X}_i\|_F^2$$

$$= \mathrm{tr}[\mathbf{X}_i \log \mathbf{X}_i] + \mathrm{tr}[(\mathbf{P}_i - \mathbf{X}_i) \log \mathbf{X}_i] + \mathrm{tr}[\mathbf{P}_i - \mathbf{X}_i] + \frac{1}{2}\|\mathbf{P}_i - \mathbf{X}_i\|_F^2$$

$$= \mathrm{tr}[\mathbf{P}_i \log \mathbf{X}_i] + \frac{1}{2}\|\mathbf{P}_i - \mathbf{X}_i\|_F^2 \tag{A.12}$$

where we used that $\mathrm{tr}[\mathbf{P}_i - \mathbf{X}_i] = 0$. Hence, by reordering, we automatically get that

$$D_i(\mathbf{P}_i, \mathbf{X}_i) \geq \frac{1}{2}\|\mathbf{P}_i - \mathbf{X}_i\|_F^2 \tag{A.13}$$

Therefore, we have:

$$D(\mathbf{P}, \mathbf{X}) \geq \frac{1}{2}\sum_{i=1}^{N}\|\mathbf{P}_i - \mathbf{X}_i\|_F^2 = \frac{1}{2}\|\mathbf{P} - \mathbf{X}\|_F^2 \tag{A.14}$$

and the proof is completed. ∎

## B  Omitted proofs from Section 4

In this appendix, we develop the basic scaffolding required for the estimators (2PE) and (1PE). We begin with the construction of the estimators' sampling basis, as encoded in Proposition 1, which we restate below for convenience:

**Proposition 1.** *Let* $\mathbf{E}_j$ *be defined as* $\mathbf{E}_j = \frac{1}{\sqrt{j(j+1)}}\left(\Delta_{11} + \cdots + \Delta_{jj} - j\Delta_{j+1,j+1}\right)$ *for* $j = 1, \ldots, d-1$. *Then, the set* $\mathcal{E} = \left\{\{\mathbf{E}_j\}_{j=1}^{d-1}, \{\mathbf{e}_{k\ell}\}_{k<\ell}, \{\tilde{\mathbf{e}}_{k\ell}\}_{k<\ell}\right\}$ *is an orthonormal basis of* $\mathcal{Z}$.

*Proof.* First of all, note that

$$\Delta_{k\ell}\Delta_{mn} = \begin{cases} 0 & \text{if } \ell \neq m \\ \Delta_{kn} & \text{if } \ell = m \end{cases} \tag{B.1}$$

**Unit norm.**    To begin with, we will show that all elements in $\mathcal{E}$ have unit norm. Indeed, we have:

- For $j = 1, \ldots, d - 1$, we have:

$$\|\mathbf{E}_j\|_F^2 = \text{tr}[\mathbf{E}_j^\dagger \mathbf{E}_j] = \frac{1}{j(j+1)} \, \text{tr}\left[\left(\sum_{k=1}^{j} \boldsymbol{\Delta}_{kk} - j\boldsymbol{\Delta}_{(j+1)(j+1)}\right)\left(\sum_{k=1}^{j} \boldsymbol{\Delta}_{kk} - j\boldsymbol{\Delta}_{(j+1)(j+1)}\right)\right]$$

$$= \frac{1}{j(j+1)} \, \text{tr}\left[\left(\sum_{k=1}^{j} \boldsymbol{\Delta}_{kk} + j^2 \boldsymbol{\Delta}_{(j+1)(j+1)}\right)\right] = \frac{1}{j(j+1)}(j + j^2) = 1 \quad \text{(B.2)}$$

- For $k < \ell$, we have:

$$\|\mathbf{e}_{k\ell}\|_F^2 = \text{tr}[\mathbf{e}_{k\ell}^\dagger \mathbf{e}_{k\ell}] = \text{tr}\left[\left(\frac{1}{\sqrt{2}}\boldsymbol{\Delta}_{\ell k} + \frac{1}{\sqrt{2}}\boldsymbol{\Delta}_{k\ell}\right)\left(\frac{1}{\sqrt{2}}\boldsymbol{\Delta}_{k\ell} + \frac{1}{\sqrt{2}}\boldsymbol{\Delta}_{\ell k}\right)\right]$$

$$= \text{tr}\left[\frac{1}{2}\boldsymbol{\Delta}_{kk} + \frac{1}{2}\boldsymbol{\Delta}_{\ell\ell}\right] = \frac{1}{2} + \frac{1}{2} = 1 \quad \text{(B.3)}$$

- For $k < \ell$, we also have:

$$\|\tilde{\mathbf{e}}_{k\ell}\|_F^2 = \text{tr}[\tilde{\mathbf{e}}_{k\ell}^\dagger \tilde{\mathbf{e}}_{k\ell}] = \text{tr}\left[\left(-\frac{i}{\sqrt{2}}\boldsymbol{\Delta}_{\ell k} + \frac{i}{\sqrt{2}}\boldsymbol{\Delta}_{k\ell}\right)\left(\frac{i}{\sqrt{2}}\boldsymbol{\Delta}_{k\ell} - \frac{i}{\sqrt{2}}\boldsymbol{\Delta}_{\ell k}\right)\right]$$

$$= \text{tr}\left[\frac{1}{2}\boldsymbol{\Delta}_{kk} + \frac{1}{2}\boldsymbol{\Delta}_{\ell\ell}\right] = \frac{1}{2} + \frac{1}{2} = 1 \quad \text{(B.4)}$$

**Orthogonality.**    Now, we will show that any two elements of $\mathcal{E}$ are orthogonal to each other.

- For $m < n$, we have:

$$\text{tr}[\mathbf{E}_m^\dagger \mathbf{E}_n] = \frac{1}{\sqrt{m(m+1)}\sqrt{n(n+1)}} \, \text{tr}\left[\left(\sum_{k=1}^{m} \boldsymbol{\Delta}_{kk} - m\boldsymbol{\Delta}_{(m+1)(m+1)}\right)\left(\sum_{k=1}^{n} \boldsymbol{\Delta}_{kk} - n\boldsymbol{\Delta}_{(n+1)(n+1)}\right)\right]$$

$$= \frac{1}{\sqrt{m(m+1)}\sqrt{n(n+1)}} \, \text{tr}\left[\left(\sum_{k=1}^{m} \boldsymbol{\Delta}_{kk} - m\boldsymbol{\Delta}_{(m+1)(m+1)}\right)\right]$$

$$= \frac{1}{\sqrt{m(m+1)}\sqrt{n(n+1)}}(m - m) = 0 \quad \text{(B.5)}$$

- For $k < \ell$, we have:

$$\text{tr}[\mathbf{e}_{k\ell}^\dagger \tilde{\mathbf{e}}_{k\ell}] = \text{tr}\left[\left(\frac{1}{\sqrt{2}}\boldsymbol{\Delta}_{\ell k} + \frac{1}{\sqrt{2}}\boldsymbol{\Delta}_{k\ell}\right)\left(\frac{i}{\sqrt{2}}\boldsymbol{\Delta}_{k\ell} - \frac{i}{\sqrt{2}}\boldsymbol{\Delta}_{\ell k}\right)\right]$$

$$= \text{tr}\left[\frac{i}{2}\boldsymbol{\Delta}_{\ell\ell} - \frac{i}{2}\boldsymbol{\Delta}_{kk}\right] = \frac{i}{2} - \frac{i}{2} = 0 \quad \text{(B.6)}$$

- For $(k, \ell) \neq (m, n)$ with $k < \ell$ and $m < n$, we have:

$$\text{tr}\left[\mathbf{e}_{k\ell}^\dagger \mathbf{e}_{mn}\right] = \text{tr}\left[\mathbf{e}_{k\ell}^\dagger \tilde{\mathbf{e}}_{mn}\right] = \text{tr}\left[\tilde{\mathbf{e}}_{k\ell}^\dagger \tilde{\mathbf{e}}_{mn}\right] = 0 \quad \text{(B.7)}$$

since all the nonzero terms in $\mathbf{e}_{k\ell}^\dagger \mathbf{e}_{mn}$, $\mathbf{e}_{k\ell}^\dagger \tilde{\mathbf{e}}_{mn}$ and $\tilde{\mathbf{e}}_{k\ell}^\dagger \tilde{\mathbf{e}}_{mn}$ are of the form $c \cdot \boldsymbol{\Delta}_{\alpha\beta}$ for some $c \in \mathbb{C}$, and $\alpha, \beta \in \{k, \ell, m, n\}$ with $\alpha \neq \beta$. Thus, $\text{tr}\left[c \cdot \boldsymbol{\Delta}_{\alpha\beta}\right] = 0$, since all the diagonal elements are equal to 0. Note that it is not possible to have $\alpha = \beta$ because this would imply that $(k, \ell) = (m, n)$.

- For $k < \ell$ and $j = 1, \ldots, d - 1$, we have:

$$\text{tr}\left[\mathbf{e}_{k\ell}^\dagger \mathbf{E}_j\right] = \text{tr}\left[\tilde{\mathbf{e}}_{k\ell}^\dagger \mathbf{E}_j\right] = 0 \quad \text{(B.8)}$$

since the non-zero terms of both $\mathbf{e}_{k\ell}^\dagger \mathbf{E}_j$ and $\tilde{\mathbf{e}}_{k\ell}^\dagger \mathbf{E}_j$ are of the form $\boldsymbol{\Delta}_{kn}, \boldsymbol{\Delta}_{\ell m}$ for $k \neq n$ and $\ell \neq m$.

We thus conclude that any two elements of $\mathcal{E}$ are orthogonal.

Finally, it is clear $\mathcal{E} \subseteq \operatorname{aff}(\mathcal{X}_0)$, since $\mathcal{E} \subseteq \mathbb{H}^d$ and $\operatorname{tr}[\mathbf{e}_{k\ell}] = \operatorname{tr}[\tilde{\mathbf{e}}_{k\ell}] = \operatorname{tr}[\mathbf{E}_j] = 0$, for $k < \ell$ and $j = 1, \ldots, d-1$. Therefore, the elements in $\mathcal{E}$ form an orthonormal basis of $\operatorname{aff}(\mathcal{X}_0)$ and $\dim(\operatorname{aff}(\mathcal{X}_0)) = d^2 - 1$. ∎

We now proceed with the construction of the precise "safety net" that guarantees that the sampling perturbation of the gradient estimator remains within the problem's feasible region. Again, for convenience, we restate the relevant result below:

**Proposition 2.** *Let* $\mathbf{R} = \frac{1}{d} \sum_{j=1}^{d} \Delta_{jj}$. *Then, for* $r = \min\left\{ \frac{1}{\sqrt{d(d-1)}}, \frac{\sqrt{2}}{d} \right\}$, *it holds that* $\mathbf{R} + r\mathbf{Z} \in \mathcal{X}$ *for any direction* $\mathbf{Z} \in \mathcal{E}^{\pm}$.

*Proof.* To begin with, it is clear that $\mathbf{R} \in \mathbb{H}^d$ and $\operatorname{tr}[\mathbf{R}] = \sum_{j=1}^{d} 1/d = 1$. Moreover, for any $u \in \mathbb{C}^d \setminus \{0\}$, we have:

$$u^\dagger \mathbf{R} u = \frac{1}{d} \sum_{j=1}^{d} |u_j|^2 > 0 \tag{B.9}$$

where $|u_j|$ is the modulus of the complex number $u_j \in \mathbb{C}$. Therefore, $\mathbf{R}$ is positive definite, i.e., lies in $\operatorname{ri}(\mathcal{X})$.

Now, we need to find $r > 0$ such that

$$\mathbf{R} + r\mathbf{Z} \in \mathcal{X} \tag{B.10}$$

for any $\mathbf{Z} \in \mathcal{E}^{\pm}$.

It is clear that for any $\mathbf{Z} \in \mathcal{E}^{\pm}$, we have $\operatorname{tr}[\mathbf{R} + r\mathbf{Z}] = \operatorname{tr}[\mathbf{R}] = 1$, since $\operatorname{tr}[\mathbf{Z}] = 0$. Hence, it remains to consider the positive semi-definite constraint. For this, we will use the following identities, for $k < \ell$:

$$u^\dagger (\Delta_{k\ell} + \Delta_{\ell k}) u = \bar{u}_k u_\ell + \bar{u}_\ell u_k = 2 \operatorname{Re}(\bar{u}_k u_\ell) \tag{B.11}$$

and

$$u^\dagger (i\Delta_{k\ell} - i\Delta_{\ell k}) u = i(\bar{u}_k u_\ell - \bar{u}_\ell u_k) = -2 \operatorname{Im}(\bar{u}_k u_\ell) \tag{B.12}$$

- For $\mathbf{Z} = \frac{1}{\sqrt{2}} (\Delta_{k\ell} + \Delta_{\ell k})$ and $u \in \mathbb{C}^d \setminus \{0\}$, and using (B.11), we have:

$$u^\dagger (\mathbf{R} + r\mathbf{Z}) u = \frac{1}{d} \sum_{j=1}^{d} |u_j|^2 + \frac{r}{\sqrt{2}} 2 \operatorname{Re}(\bar{u}_k u_\ell)$$

$$= \frac{1}{d} \sum_{j \neq k, \ell} |u_j|^2 + \frac{1}{d} \left( |u_k|^2 + |u_\ell|^2 + \frac{rd}{\sqrt{2}} 2 \operatorname{Re}(\bar{u}_k u_\ell) \right) \tag{B.13}$$

If $\operatorname{Re}(\bar{u}_k u_\ell) > 0$, we get that $u^\dagger (\mathbf{R} + r\mathbf{Z}) u > 0$, while if $\operatorname{Re}(\bar{u}_k u_\ell) \leq 0$ and $r \leq \sqrt{2}/d$:

$$u^\dagger (\mathbf{R} + r\mathbf{Z}) u \geq \frac{1}{d} \sum_{j \neq k, \ell} |u_j|^2 + \frac{1}{d} |u_k + u_\ell|^2 \geq 0 \tag{B.14}$$

Hence, for $r \leq \sqrt{2}/d$, and $\mathbf{Z} = \frac{1}{\sqrt{2}} (\Delta_{k\ell} + \Delta_{\ell k})$, we have that $u^\dagger (\mathbf{R} + r\mathbf{Z}) u \geq 0$ for all $u \in \mathbb{C}^d$.

- For $\mathbf{Z} = -\frac{1}{\sqrt{2}} (\Delta_{k\ell} + \Delta_{\ell k})$, we have

$$u^\dagger (\mathbf{R} + r\mathbf{Z}) u = \frac{1}{d} \sum_{j=1}^{d} |u_j|^2 - \frac{r}{\sqrt{2}} 2 \operatorname{Re}(\bar{u}_k u_\ell)$$

$$= \frac{1}{d} \sum_{j \neq k, \ell} |u_j|^2 + \frac{1}{d} \left( |u_k|^2 + |u_\ell|^2 - \frac{rd}{\sqrt{2}} 2 \operatorname{Re}(\bar{u}_k u_\ell) \right) \tag{B.15}$$

If $\operatorname{Re}(\bar{u}_k u_\ell) < 0$, we get that $u^\dagger (\mathbf{R} + r\mathbf{Z}) u > 0$, while if $\operatorname{Re}(\bar{u}_k u_\ell) \geq 0$ and $r \leq \sqrt{2}/d$:

$$u^\dagger (\mathbf{R} + r\mathbf{Z}) u \geq \frac{1}{d} \sum_{j \neq k, \ell} |u_j|^2 + \frac{1}{d} |u_k - u_\ell|^2 \geq 0 \tag{B.16}$$

Hence, for $r \leq \sqrt{2}/d$, and $\mathbf{Z} = -\frac{1}{\sqrt{2}} (\Delta_{k\ell} + \Delta_{\ell k})$, we have that $u^\dagger (\mathbf{R} + r\mathbf{Z}) u \geq 0$ for all $u \in \mathbb{C}^d$.

- For $\mathbf{Z} = \frac{i}{\sqrt{2}}(\Delta_{k\ell} - \Delta_{\ell k})$ and $u \in \mathbb{C}^d \setminus \{0\}$, and using (B.11), we have:

$$u^\dagger(\mathbf{R} + r\mathbf{Z})u = \frac{1}{d}\sum_{j=1}^{d}|u_j|^2 - \frac{r}{\sqrt{2}}2\text{Im}(\bar{u}_k u_\ell)$$

$$= \frac{1}{d}\sum_{j\neq k,\ell}|u_j|^2 + \frac{1}{d}\left(|u_k|^2 + |u_\ell|^2 - \frac{rd}{\sqrt{2}}2\text{Im}(\bar{u}_k u_\ell)\right) \tag{B.17}$$

If $\text{Im}(\bar{u}_k u_\ell) < 0$, we get that $u^\dagger(\mathbf{R} + r\mathbf{Z})u > 0$, while if $\text{Im}(\bar{u}_k u_\ell) \geq 0$ and $r \leq \sqrt{2}/d$:

$$u^\dagger(\mathbf{R} + r\mathbf{Z})u \geq \frac{1}{d}\sum_{j\neq k,\ell}|u_j|^2 + \frac{1}{d}|u_k + i\,u_\ell|^2 \geq 0 \tag{B.18}$$

Hence, for $r \leq \sqrt{2}/2d$, and $\mathbf{Z} = \frac{i}{\sqrt{2}}(\Delta_{k\ell} - \Delta_{\ell k})$, we have that $u^\dagger(\mathbf{R} + r\mathbf{Z})u \geq 0$ for all $u \in \mathbb{C}^d$.

- For $\mathbf{Z} = -\frac{i}{\sqrt{2}}(\Delta_{k\ell} - \Delta_{\ell k})$ and $u \in \mathbb{C}^d \setminus \{0\}$, and using (B.11), we have:

$$u^\dagger(\mathbf{R} + r\mathbf{Z})u = \frac{1}{d}\sum_{j=1}^{d}|u_j|^2 + \frac{r}{\sqrt{2}}2\text{Im}(\bar{u}_k u_\ell)$$

$$= \frac{1}{d}\sum_{j\neq k,\ell}|u_j|^2 + \frac{1}{d}\left(|u_k|^2 + |u_\ell|^2 + \frac{rd}{\sqrt{2}}2\text{Im}(\bar{u}_k u_\ell)\right) \tag{B.19}$$

If $\text{Im}(\bar{u}_k u_\ell) > 0$, we get that $u^\dagger(\mathbf{R} + r\mathbf{Z})u > 0$, while if $\text{Im}(\bar{u}_k u_\ell) \leq 0$ and $r \leq \sqrt{2}/d$:

$$u^\dagger(\mathbf{R} + r\mathbf{Z})u \geq \frac{1}{d}\sum_{j\neq k,\ell}|u_j|^2 + \frac{1}{d}|u_k - i\,u_\ell|^2 \tag{B.20}$$

Hence, for $r \leq \sqrt{2}/2d$, and $\mathbf{Z} = -\frac{i}{\sqrt{2}}(\Delta_{k\ell} - \Delta_{\ell k})$, we have that $u^\dagger(\mathbf{R} + r\mathbf{Z})u \geq 0$ for all $u \in \mathbb{C}^d$.

- For $\mathbf{Z} = \frac{1}{\sqrt{j(j+1)}}\left(\Delta_{11} + \cdots + \Delta_{jj} - j\Delta_{(j+1)(j+1)}\right)$, we have:

$$u^\dagger(\mathbf{R} + r\mathbf{Z})u = \frac{1}{d}\sum_{k=1}^{d}|u_k|^2 + \frac{r}{\sqrt{j(j+1)}}\sum_{k=1}^{j}|u_k|^2 - \frac{jr}{\sqrt{j(j+1)}}|u_{j+1}|^2$$

$$= \frac{1}{d}\sum_{k\neq j+1}|u_k|^2 + \frac{r}{\sqrt{j(j+1)}}\sum_{k=1}^{j}|u_k|^2 + \left(\frac{1}{d} - \frac{jr}{\sqrt{j(j+1)}}\right)|u_{j+1}|^2 \tag{B.21}$$

Thus, we need to ensure that

$$\frac{1}{d} - \frac{jr}{\sqrt{j(j+1)}} \geq 0 \tag{B.22}$$

for all $j = 1, \ldots, d-1$. Because the function $x \mapsto \sqrt{x(x+1)}/x$ is decreasing, it obtains the smallest value from $x = d - 1$. Therefore, for $r \leq 1/\sqrt{d(d-1)}$, we readily obtain that $u^\dagger(\mathbf{R} + r\mathbf{Z})u \geq 0$ for all $u \in \mathbb{C}^d$.

- For $\mathbf{Z} = -\frac{1}{\sqrt{j(j+1)}}\left(\Delta_{11} + \cdots + \Delta_{jj} - j\Delta_{(j+1)(j+1)}\right)$, we have:

$$u^\dagger(\mathbf{R} + r\mathbf{Z})u = \frac{1}{d}\sum_{k=1}^{d}|u_k|^2 - \frac{r}{\sqrt{j(j+1)}}\sum_{k=1}^{j}|u_k|^2 + \frac{jr}{\sqrt{j(j+1)}}|u_{j+1}|^2$$

$$= \left(\frac{1}{d} - \frac{r}{\sqrt{j(j+1)}}\right)\sum_{k=1}^{j}|u_k|^2 + \frac{1}{d}\sum_{k=j+1}^{d}|u_k|^2 + \frac{jr}{\sqrt{j(j+1)}}|u_{j+1}|^2 \tag{B.23}$$

Thus, we need to ensure that

$$\frac{1}{d} - \frac{r}{\sqrt{j(j+1)}} \geq 0 \tag{B.24}$$

for all $j = 1, \ldots, d - 1$. Because it holds that

$$\frac{1}{d} - \frac{r}{\sqrt{j(j+1)}} \geq \frac{1}{d} - \frac{jr}{\sqrt{j(j+1)}} \tag{B.25}$$

we obtain the inequality for free by the previous case, i.e., for $r \leq 1/\sqrt{d(d-1)}$.

Therefore, for

$$r = \min\left\{\frac{1}{\sqrt{d(d-1)}}, \frac{\sqrt{2}}{d}\right\} \tag{B.26}$$

we readily obtain that $u^\dagger(\mathbf{R} + r\mathbf{Z})u \geq 0$ for all $u \in \mathbb{C}^d$, and our proof is complete. ∎

# C Omitted proofs from Section 5

Our aim in this appendix will be to prove the basic guarantees of (3MW) with payoff-based feedback. The structure of this appendix shadows that of Section 5 and is broken into two parts, depending on the specific type of input available to the players. The only point of departure is the energy inequality of Lemma 1, which is common to both algorithms, and which we restate and prove below:

**Lemma 1.** *Fix some $\mathbf{P} \in \mathcal{X}$, and let $\mathbf{X}_t, \mathbf{X}_{t+1}$ be two successive iterates of* (3MW)*, without any assumptions for the input sequence $\hat{\mathbf{V}}_t$. We then have*

$$D(\mathbf{P}, \mathbf{X}_{t+1}) \leq D(\mathbf{P}, \mathbf{X}_t) + \gamma_t \operatorname{tr}[\hat{\mathbf{V}}_t(\mathbf{X}_t - \mathbf{P})] + \frac{\gamma_t^2}{2}\|\hat{\mathbf{V}}_t\|_F^2. \tag{18}$$

*Proof.* By the definition of $D$, it is easy to see that for $\mathbf{P} \in \mathcal{X}$ and $\mathbf{X}, \mathbf{X}' \in \operatorname{ri}(\mathcal{X})$, we have

$$D(\mathbf{P}, \mathbf{X}') = D(\mathbf{P}, \mathbf{X}) + D(\mathbf{X}, \mathbf{X}') + \operatorname{tr}[(\log \mathbf{X}' - \log \mathbf{X})(\mathbf{X} - \mathbf{P})] \tag{C.1}$$

Since $\nabla h(\mathbf{X}) = \log \mathbf{X} + \mathbf{I}$, the above equality can be written as:

$$D(\mathbf{P}, \mathbf{X}') = D(\mathbf{P}, \mathbf{X}) + D(\mathbf{X}, \mathbf{X}') + \operatorname{tr}[(\nabla h(\mathbf{X}') - \nabla h(\mathbf{X}))(\mathbf{X} - \mathbf{P})] \tag{C.2}$$

Setting $\mathbf{X}$ as $\mathbf{X}_{t+1}$, and $\mathbf{X}'$ as $\mathbf{X}_t$, and invoking the easily verifiable fact that $\nabla h(\mathbf{X}_{t+1}) - \nabla h(\mathbf{X}_t) = \gamma_t \hat{\mathbf{V}}_t$, we get

$$D(\mathbf{P}, \mathbf{X}_t) = D(\mathbf{P}, \mathbf{X}_{t+1}) + D(\mathbf{X}_{t+1}, \mathbf{X}_t) - \gamma_t \operatorname{tr}[\hat{\mathbf{V}}_t(\mathbf{X}_{t+1} - \mathbf{P})] \tag{C.3}$$

and hence:

$$\begin{aligned}
D(\mathbf{P}, \mathbf{X}_{t+1}) &= D(\mathbf{P}, \mathbf{X}_t) - D(\mathbf{X}_{t+1}, \mathbf{X}_t) + \gamma_t \operatorname{tr}[\hat{\mathbf{V}}_t(\mathbf{X}_{t+1} - \mathbf{P})] \\
&\leq D(\mathbf{P}, \mathbf{X}_t) - \frac{1}{2}\|\mathbf{X}_{t+1} - \mathbf{X}_t\|_F^2 + \gamma_t \operatorname{tr}[\hat{\mathbf{V}}_t(\mathbf{X}_{t+1} - \mathbf{P})] \\
&= D(\mathbf{P}, \mathbf{X}_t) - \frac{1}{2}\|\mathbf{X}_{t+1} - \mathbf{X}_t\|_F^2 + \gamma_t \operatorname{tr}[\hat{\mathbf{V}}_t(\mathbf{X}_t - \mathbf{P})] + \gamma_t \operatorname{tr}[\hat{\mathbf{V}}_t(\mathbf{X}_{t+1} - \mathbf{X}_t)] \\
&\leq D(\mathbf{P}, \mathbf{X}_t) + \gamma_t \operatorname{tr}[\hat{\mathbf{V}}_t(\mathbf{X}_t - \mathbf{P})] + \frac{\gamma_t^2}{2}\|\hat{\mathbf{V}}_t\|_F^2 \tag{C.4}
\end{aligned}$$

where the first inequality holds due to Lemma A.2, and in the last step we used that $\|\cdot\|_F$ is an inner-product norm on $\mathcal{X}$, so

$$\frac{1}{2}\|\mathbf{X}_{t+1} - \mathbf{X}_t\|_F^2 + \frac{\gamma_t^2}{2}\|\hat{\mathbf{V}}_t\|_F^2 \geq \gamma_t \operatorname{tr}[\hat{\mathbf{V}}_t(\mathbf{X}_{t+1} - \mathbf{X}_t)] \tag{C.5}$$

This concludes our proof. ∎

With this template inequality in hand, we proceed with the guarantees of (3MW) in the next sections.

**C.1. Learning with mixed payoff observations.** We begin with the statistics of the 2-point sampler (2PE), which we restate below:

**Proposition 3.** *The estimator* (2PE) *enjoys the conditional bounds*

$$(i) \quad \big\| \mathbb{E}[\hat{\mathbf{V}}_t \,|\, \mathcal{F}_t] - \mathbf{V}(\mathbf{X}_t) \big\|_F \leq 4DL\delta_t \quad and \quad (ii) \quad \mathbb{E}\big[ \|\hat{\mathbf{V}}_t\|_F^2 \,\big|\, \mathcal{F}_t \big] \leq 16D^2G^2 \qquad (19)$$

*Proof.* We prove each part separately.

*(i)* Let $\Xi_{i,t}^{(+)}$ and $\Xi_{i,t}^{(-)}$ be defined for all players $i \in \{1, 2\}$ as

$$\Xi_{i,t}^{(+)} = (\mathbf{X}_{i,t}^{(\delta)} + s_{i,t}\delta_t \mathbf{Z}_{i,t}) - \mathbf{X}_{i,t}$$
$$= s_{i,t}\delta_t \mathbf{Z}_{i,t} + \frac{\delta_t}{r_i}(\mathbf{R}_i - \mathbf{X}_{i,t}) = \delta_t \left[ s_{i,t}\mathbf{Z}_{i,t} + \frac{1}{r_i}(\mathbf{R}_i - \mathbf{X}_{i,t}) \right] \qquad (C.6)$$

and

$$\Xi_{i,t}^{(-)} = (\mathbf{X}_{i,t}^{(\delta)} - s_{i,t}\delta_t \mathbf{Z}_{i,t}) - \mathbf{X}_{i,t} = \delta_t \left[ -s_{i,t}\mathbf{Z}_{i,t} + \frac{1}{r_i}(\mathbf{R}_i - \mathbf{X}_{i,t}) \right] \qquad (C.7)$$

Taking a first-order Taylor expansion of $u_i$, we obtain:

$$u_i(\mathbf{X}_t^{(\delta)} + s_t\delta_t \mathbf{Z}_t) = u_i(\mathbf{X}_t) + \sum_{j \in \mathcal{N}} \mathrm{tr}\left[ \nabla_{\mathbf{X}_j^\top} u_i(\mathbf{X}_t)^\dagger \Xi_{j,t}^{(+)} \right] + R_2(\Xi_t^{(+)}) \qquad (C.8a)$$

and

$$u_i(\mathbf{X}_t^{(\delta)} - s_t\delta_t \mathbf{Z}_t) = u_i(\mathbf{X}_t) + \sum_{j \in \mathcal{N}} \mathrm{tr}\left[ \nabla_{\mathbf{X}_j^\top} u_i(\mathbf{X}_t)^\dagger \Xi_{j,t}^{(-)} \right] + R_2(\Xi_t^{(-)}) \qquad (C.8b)$$

where $R_2(\cdot)$ is the 2nd order Taylor remainder. Now, for $j \neq i \in \mathcal{N}$, since $s_{i,t}$ is zero-mean and independent of any other process:

$$\mathbb{E}\left[ \mathrm{tr}\left[ \nabla_{\mathbf{X}_j^\top} u_i(\mathbf{X}_t)^\dagger (\Xi_{j,t}^{(+)} - \Xi_{j,t}^{(-)}) \right] s_{i,t}\mathbf{Z}_{i,t} \,\Big|\, \mathcal{F}_t \right] = 0 \qquad (C.9)$$

and using that $\Xi_{i,t}^{(+)} - \Xi_{i,t}^{(-)} = 2s_{i,t}\delta_t \mathbf{Z}_{i,t}$, we have:

$$\mathbb{E}\left[ \mathrm{tr}\left[ \nabla_{\mathbf{X}_i^\top} u_i(\mathbf{X}_t)^\dagger (\Xi_{i,t}^{(+)} - \Xi_{i,t}^{(-)}) \right] s_{i,t}\mathbf{Z}_{i,t} \,\Big|\, \mathcal{F}_t \right] = \mathbb{E}\left[ \mathrm{tr}\left[ \mathbf{V}_i(\mathbf{X}_t)^\dagger (2s_{i,t}\delta_t \mathbf{Z}_{i,t}) \right] s_{i,t}\mathbf{Z}_{i,t} \,|\, \mathcal{F}_t \right]$$
$$= 2\delta_t \, \mathbb{E}\left[ \mathrm{tr}\left[ \mathbf{V}_i(\mathbf{X}_t)^\dagger \mathbf{Z}_{i,t} \right] s_{i,t}^2 \mathbf{Z}_{i,t} \,|\, \mathcal{F}_t \right]$$
$$= 2\delta_t \, \mathbb{E}\left[ \mathrm{tr}\left[ \mathbf{V}_i(\mathbf{X}_t)^\dagger \mathbf{Z}_{i,t} \right] \mathbf{Z}_{i,t} \,|\, \mathcal{F}_t \right]$$
$$= \frac{2\delta_t}{D_i} \sum_{W \in \mathcal{E}_i} \mathrm{tr}\left[ \mathbf{V}_i(\mathbf{X}_t)^\dagger W \right] W$$
$$= \frac{2\delta_t}{D_i} \mathrm{proj}_{\mathcal{E}_i}(\mathbf{V}_i(\mathbf{X}_t)) = \frac{2\delta_t}{D_i} \mathbf{V}_i(\mathbf{X}_t) \quad (C.10)$$

where in the last step, with a slight abuse of notation, we identify $\mathrm{proj}_{\mathcal{E}_i}(\mathbf{V}_i(\mathbf{X}_t))$ with $\mathbf{V}_i(\mathbf{X}_t)$. The reason for this is that we apply the differential operator $\mathbf{V}_i(\mathbf{X}_t)$ only on elements of $\mathcal{X}_i$, and thus, we can ignore the component of $\mathbf{V}_i(\mathbf{X}_t)$ that is orthogonal to $\mathrm{span}(\mathcal{E}_i)$.

Moreover, we have that

$$|R_2(\Xi_t^{(+)})| \leq \frac{L}{2} \|\Xi_t^{(+)}\|_F^2 \leq L\delta_t^2 \qquad (C.11)$$

and similarly, we get the same bound for $|R_2(\Xi_t^{(-)})|$. Therefore, in light of the above, we obtain the bound:

$$\|\mathbb{E}[\hat{\mathbf{V}}_{i,t} \,|\, \mathcal{F}_t] - \mathbf{V}_i(\mathbf{X}_t)\|_F \leq \frac{1}{2} D_i L \delta_t \qquad (C.12a)$$

and, hence

$$\|\mathbb{E}[\hat{\mathbf{V}}_t \,|\, \mathcal{F}_t] - \mathbf{V}(\mathbf{X}_t)\|_F \leq \frac{\sqrt{2}}{2} DL\delta_t \qquad (C.12b)$$

*(ii)* By the definition of $\hat{\mathbf{V}}_{i,t}$, we have:

$$\|\hat{\mathbf{V}}_{i,t}\|_F = \frac{D_i}{2\delta_t}\left|u_i(\mathbf{X}_t^{(\delta)} + s_t\,\delta_t\,\mathbf{Z}_t) - u_i(\mathbf{X}_t^{(\delta)} - s_t\,\delta_t\,\mathbf{Z}_t)\right|\|s_{i,t}\mathbf{Z}_{i,t}\|_F$$

$$\leq \frac{D_i}{2\delta_t}G\|2s_t\,\delta_t\,\mathbf{Z}_t\|_F \leq \sqrt{2}D_iG \tag{C.13}$$

and therefore, we readily obtain that:

$$\mathbb{E}\left[\|\hat{\mathbf{V}}_{i,t}\|_F^2\,\big|\,\mathcal{F}_t\right] \leq 2D_i^2G^2 \tag{C.14}$$

so

$$\mathbb{E}\left[\|\hat{\mathbf{V}}_t\|_F^2\,\big|\,\mathcal{F}_t\right] \leq 4D^2G^2 \tag{C.15}$$

and our proof is complete. ∎

With all these technical elements in place, we are finally in a position to prove our convergence result for (3MW) run with 2-point gradient estimators. As before, we restate our result below for convenience:

**Theorem 2.** *Suppose that each player of a 2-player zero-sum game $\mathcal{Q}$ follows* (3MW) *for $T$ epochs with learning rate $\gamma$, sampling radius $\delta$, and gradient estimates provided by* (2PE)*. Then the players' empirical frequency of play enjoys the duality gap guarantee*

$$\mathbb{E}\left[\mathrm{Gap}_{\mathcal{L}}(\bar{\mathbf{X}}_T)\right] \leq \frac{H}{\gamma T} + 8D^2G^2\gamma + 16DL\delta \tag{15}$$

*where $H = \log(d_1 d_2)$. In particular, for $\gamma = (DG)^{-1}\sqrt{H/(8T)}$ and $\delta = (G/L)\sqrt{H/(8T)}$, the players enjoy the equilibrium convergence guarantee*

$$\mathbb{E}\left[\mathrm{Gap}_{\mathcal{L}}(\bar{\mathbf{X}}_T)\right] \leq 8DG\sqrt{2H/T}. \tag{16}$$

*Proof.* Let $\mathbf{X}^* \in \mathcal{X}$ be a NE point. By Lemma 1 for $\mathbf{P} = \mathbf{X}^*$, and setting $F_t := D(\mathbf{X}^*, \mathbf{X}_t)$ for all $t = 1, 2 \ldots$, we have

$$F_{t+1} \leq F_t + \gamma_t\,\mathrm{tr}[\hat{\mathbf{V}}_t^\dagger(\mathbf{X}_t - \mathbf{X}^*)] + \frac{\gamma_t^2}{2}\|\hat{\mathbf{V}}_t\|_F^2 \tag{C.16}$$

or, equivalently

$$\mathrm{tr}[\hat{\mathbf{V}}_t^\dagger(\mathbf{X}^* - \mathbf{X}_t)] \leq \frac{1}{\gamma_t}(F_t - F_{t+1}) + \frac{\gamma_t}{2}\|\hat{\mathbf{V}}_t\|_F^2 \tag{C.17}$$

Summing over the whole sequence $t = 1, \ldots, T$, we get:

$$\sum_{t=1}^{T}\mathrm{tr}[\hat{\mathbf{V}}_t^\dagger(\mathbf{X}^* - \mathbf{X}_t)] \leq \sum_{t=1}^{T}\frac{1}{\gamma_t}(F_t - F_{t+1}) + \frac{1}{2}\sum_{t=1}^{T}\gamma_t\|\hat{\mathbf{V}}_t\|_F^2 \tag{C.18}$$

which can be rewritten by setting $\gamma_0 = \infty$, as:

$$\sum_{t=1}^{T}\mathrm{tr}[\hat{\mathbf{V}}_t^\dagger(\mathbf{X}^* - \mathbf{X}_t)] \leq \sum_{t=1}^{T}F_t\left(\frac{1}{\gamma_t} - \frac{1}{\gamma_{t-1}}\right) + \frac{1}{2}\sum_{t=1}^{T}\gamma_t\|\hat{\mathbf{V}}_t\|_F^2 \tag{C.19}$$

Decomposing $\hat{\mathbf{V}}_t$ as

$$\hat{\mathbf{V}}_t = \mathbf{V}(\mathbf{X}_t) + b_t + U_t \tag{C.20}$$

with

*(i)* $b_t = \mathbb{E}\left[\hat{\mathbf{V}}_t\,\big|\,\mathcal{F}_t\right] - \mathbf{V}(\mathbf{X}_t)$

*(ii)* $U_t = \hat{\mathbf{V}}_t - \mathbb{E}\left[\hat{\mathbf{V}}_t\,\big|\,\mathcal{F}_t\right]$

equation (D.18) becomes:

$$\sum_{t=1}^{T} \text{tr}[\mathbf{V}(\mathbf{X}_t)^\dagger(\mathbf{X}^* - \mathbf{X}_t)] \leq \sum_{t=1}^{T} F_t\left(\frac{1}{\gamma_t} - \frac{1}{\gamma_{t-1}}\right) + \frac{1}{2}\sum_{t=1}^{T} \gamma_t \|\hat{\mathbf{V}}_t\|_F^2$$
$$+ \sum_{t=1}^{T} \text{tr}[b_t^\dagger(\mathbf{X}_t - \mathbf{X}^*)] + \sum_{t=1}^{T} \text{tr}[U_t^\dagger(\mathbf{X}_t - \mathbf{X}^*)]$$
$$\leq \sum_{t=1}^{T} F_t\left(\frac{1}{\gamma_t} - \frac{1}{\gamma_{t-1}}\right) + \frac{1}{2}\sum_{t=1}^{T} \gamma_t \|\hat{\mathbf{V}}_t\|_F^2$$
$$+ 4\sum_{t=1}^{T} \|b_t\|_F + \sum_{t=1}^{T} \text{tr}[U_t^\dagger(\mathbf{X}_t - \mathbf{X}^*)] \tag{C.21}$$

The left-hand side (LHS) of (D.20) gives:

$$\sum_{t=1}^{T} \text{tr}[\mathbf{V}(\mathbf{X}_t)^\dagger(\mathbf{X}^* - \mathbf{X}_t)] = \sum_{t=1}^{T} \text{tr}[\mathbf{V}_1(\mathbf{X}_t)^\dagger(\mathbf{X}_1^* - \mathbf{X}_{1,t})] + \sum_{t=1}^{T} \text{tr}[\mathbf{V}_2(\mathbf{X}_t)^\dagger(\mathbf{X}_2^* - \mathbf{X}_{2,t})]$$
$$= \sum_{t=1}^{T}\left(u_1(\mathbf{X}_1^*, \mathbf{X}_{2,t}) - u_1(\mathbf{X}_t)\right) + \sum_{t=1}^{T}\left(u_2(\mathbf{X}_{1,t}, \mathbf{X}_2^*) - u_2(\mathbf{X}_t)\right)$$
$$= \sum_{t=1}^{T}\left(\mathcal{L}(\mathbf{X}_1^*, \mathbf{X}_{2,t}) - \mathcal{L}(\mathbf{X}_{1,t}, \mathbf{X}_2^*)\right) \tag{C.22}$$

Hence, dividing by $T$, we get:

$$\mathcal{L}(\mathbf{X}_1^*, \bar{\mathbf{X}}_{2,T}) - \mathcal{L}(\bar{\mathbf{X}}_{1,T}, \mathbf{X}_2^*) \leq \frac{1}{T}\sum_{t=1}^{T} \text{tr}[\mathbf{V}(\mathbf{X}_t)^\dagger(\mathbf{X}^* - \mathbf{X}_t)] \tag{C.23}$$

or, equivalently,

$$\text{Gap}_{\mathcal{L}}(\bar{\mathbf{X}}_T) \leq \frac{1}{T}\sum_{t=1}^{T} \text{tr}[\mathbf{V}(\mathbf{X}_t)^\dagger(\mathbf{X}^* - \mathbf{X}_t)]$$
$$\leq \frac{1}{T}\sum_{t=1}^{T} F_t\left(\frac{1}{\gamma_t} - \frac{1}{\gamma_{t-1}}\right) + \frac{1}{2T}\sum_{t=1}^{T} \gamma_t \|\hat{\mathbf{V}}_t\|_F^2$$
$$+ \frac{4}{T}\sum_{t=1}^{T} \|b_t\|_F + \frac{1}{T}\sum_{t=1}^{T} \text{tr}[U_t^\dagger(\mathbf{X}_t - \mathbf{X}^*)] \tag{C.24}$$

Now, we focus on the right-hand side (RHS) of (D.20). Specifically, we have:

$$\mathbb{E}[\text{tr}[U_t^\dagger(\mathbf{X}_t - \mathbf{X}^*)]] = \mathbb{E}[\mathbb{E}[\text{tr}[U_t^\dagger(\mathbf{X}_t - \mathbf{X}^*)] \mid \mathcal{F}_t]] = 0 \tag{C.25}$$

since $\mathbf{X}_t$ is $\mathcal{F}_t$-measurable and $\mathbb{E}[U_t \mid \mathcal{F}_t] = 0$.

Moreover, by Proposition 3, we have:

$$\|b_{i,t}\|_F = \left\|\mathbb{E}[\hat{\mathbf{V}}_{i,t} \mid \mathcal{F}_t] - \mathbf{V}_i(\mathbf{X}_t)\right\|_F \leq 2D_i L \delta_t \tag{C.26a}$$

and

$$\mathbb{E}\left[\|\hat{\mathbf{V}}_{i,t}\|_F^2\right] = \mathbb{E}\left[\mathbb{E}\left[\|\hat{\mathbf{V}}_{i,t}\|_F^2 \mid \mathcal{F}_t\right]\right] \leq 4D_i^2 G^2 \tag{C.26b}$$

Hence, taking expectation in (D.20), we obtain:

$$\mathbb{E}\left[\text{Gap}_{\mathcal{L}}(\bar{\mathbf{X}}_T)\right] \leq \frac{1}{T}\sum_{t=1}^{T} \mathbb{E}[F_t]\left(\frac{1}{\gamma_t} - \frac{1}{\gamma_{t-1}}\right) + \frac{1}{2T}\sum_{t=1}^{T} \gamma_t \,\mathbb{E}[\|\hat{\mathbf{V}}_t\|_F^2] + \frac{4}{T}\sum_{t=1}^{T} \mathbb{E}[\|b_t\|_F]$$
$$\leq \frac{1}{T}\sum_{t=1}^{T} \mathbb{E}[F_t]\left(\frac{1}{\gamma_t} - \frac{1}{\gamma_{t-1}}\right) + \frac{8D^2G^2}{T}\sum_{t=1}^{T} \gamma_t + \frac{16DL}{T}\sum_{t=1}^{T} \delta_t \tag{C.27}$$

Setting $\gamma_t = \gamma$ and $\delta_t = \delta$, we obtain:

$$\mathbb{E}\big[\mathrm{Gap}_{\mathcal{L}}(\bar{\mathbf{X}}_T)\big] \leq \frac{F_1}{\gamma T} + 8D^2 G^2 \gamma + 16DL\delta \tag{C.28}$$

and finally, noting that

$$F_1 = D(\mathbf{X}^*, \mathbf{X}_1) \leq \log(d_1 d_2) \tag{C.29}$$

we get:

$$\mathbb{E}\big[\mathrm{Gap}_{\mathcal{L}}(\bar{\mathbf{X}}_T)\big] \leq \frac{H}{\gamma T} + 8D^2 G^2 \gamma + 16DL\delta \tag{C.30}$$

for $H = \log(d_1 d_2)$. Hence, after tuning $\gamma$ to optimize this last expression, our result follows by setting $\gamma = \sqrt{\frac{H}{8TD^2 G^2}}$ and $\delta = \sqrt{\frac{G^2 H}{8L^2 T}}$. $\blacksquare$

**C.2. Learning with bandit feedback.** We now proceed with the more arduous task of proving the bona fide, bandit guarantees of (3MW) with 1-point, stochastic, payoff-based feedback. The key difference with our previous analysis lies in the different statistical properties of the 1-point estimator (1PE). The relevant result that we will need is restated below:

**Proposition 4.** *The estimator* (1PE) *enjoys the conditional bounds*

$$(i) \quad \|\mathbb{E}[\hat{\mathbf{V}}_t \mid \mathcal{F}_t] - \mathbf{V}(\mathbf{X}_t)\|_F \leq 4DL\delta_t \quad and \quad (ii) \quad \mathbb{E}[\|\hat{\mathbf{V}}_t\|_F^2 \mid \mathcal{F}_t] \leq 4D^2 B^2 / \delta_t^2. \tag{20}$$

*Proof.* We prove each part separately.

*(i)* Let $\Xi_{i,t}$ be defined for all players $i \in \mathcal{N}$:

$$\Xi_{i,t} = \mathbf{X}_{i,t}^{(\delta)} - \mathbf{X}_{i,t} = \delta_t \mathbf{Z}_{i,t} + \frac{\delta_t}{r_i}(\mathbf{R}_i - \mathbf{X}_{i,t}) = \delta_t \left[ \mathbf{Z}_{i,t} + \frac{1}{r_i}(\mathbf{R}_i - \mathbf{X}_{i,t}) \right] \tag{C.31}$$

Taking a first-order Taylor expansion of $u_i$, we obtain:

$$u_i(\mathbf{X}_t^{(\delta)} + \delta_t \mathbf{Z}_t) = u_i(\mathbf{X}_t) + \sum_{j \in \mathcal{N}} \mathrm{tr}\left[ \nabla_{\mathbf{X}_j^\top} u_i(\mathbf{X}_t)^\dagger \Xi_{j,t} \right] + R_2(\Xi_t) \tag{C.32}$$

Since $\mathbb{E}[U_i(\omega_t) \mid \mathcal{F}_t, \mathbf{Z}_t] = u(\mathbf{X}_t^{(\delta)} + \delta_t \mathbf{Z}_t)$, combining it with (D.3), we readily get:

$$\mathbb{E}[\hat{\mathbf{V}}_{i,t} \mid \mathcal{F}_t, \mathbf{Z}_t] = \frac{D_i}{\delta_t} u_i(\mathbf{X}_t^{(\delta)} + \delta_t \mathbf{Z}_t) \, \mathbf{Z}_{i,t} \tag{C.33}$$

$$= \frac{D_i}{\delta_t} u_i(\mathbf{X}_t) \mathbf{Z}_{i,t} + \frac{D_i}{\delta_t} \sum_{j \in \mathcal{N}} \mathrm{tr}\left[ \nabla_{\mathbf{X}_j^\top} u_i(\mathbf{X}_t)^\dagger \Xi_{j,t} \right] \mathbf{Z}_{i,t} + \frac{D_i}{\delta_t} R_2(\Xi_t) \mathbf{Z}_{i,t} \tag{C.34}$$

Now, because $\mathbb{E}[\mathbf{Z}_{i,t} \mid \mathcal{F}_t] = 0$ and $\mathbf{Z}_{i,t}$ is sampled independent of any other process, we have:

$$\mathbb{E}[u_i(\mathbf{X}_t) \mathbf{Z}_{i,t} \mid \mathcal{F}_t] = u_i(\mathbf{X}_t) \, \mathbb{E}[\mathbf{Z}_{i,t} \mid \mathcal{F}_t] = 0 \tag{C.35}$$

and for $j \neq i \in \mathcal{N}$:

$$\mathbb{E}\left[ \mathrm{tr}\left[ \nabla_{\mathbf{X}_j^\top} u_i(\mathbf{X}_t)^\dagger \Xi_{j,t} \right] \mathbf{Z}_{i,t} \,\Big|\, \mathcal{F}_t \right] = 0 \tag{C.36}$$

Therefore, we obtain:

$$\mathbb{E}\left[ \sum_{j \in \mathcal{N}} \mathrm{tr}\left[ \nabla_{\mathbf{X}_j^\top} u_i(\mathbf{X}_t)^\dagger \Xi_{j,t} \right] \mathbf{Z}_{i,t} \mid \mathcal{F}_t \right] = \mathbb{E}\left[ \mathrm{tr}\left[ \mathbf{V}_i(\mathbf{X}_t)^\dagger \Xi_{i,t} \right] \mathbf{Z}_{i,t} \mid \mathcal{F}_t \right]$$

$$= \delta_t \, \mathbb{E}\left[ \mathrm{tr}\left[ \mathbf{V}_i(\mathbf{X}_t)^\dagger \mathbf{Z}_{i,t} \right] \mathbf{Z}_{i,t} \mid \mathcal{F}_t \right]$$

$$= \frac{\delta_t}{D_i} \sum_{W \in \mathcal{E}_i} \mathrm{tr}\left[ \mathbf{V}_i(\mathbf{X}_t)^\dagger W \right] W$$

$$= \frac{\delta_t}{D_i} \mathrm{proj}_{\mathcal{E}_i}(\mathbf{V}_i(\mathbf{X}_t)) = \frac{\delta_t}{D_i} \mathbf{V}_i(\mathbf{X}_t) \tag{C.37}$$

where in the last step, we identify $\text{proj}_{\mathcal{E}_i}(\mathbf{V}_i(\mathbf{X}_t))$ with $\mathbf{V}_i(\mathbf{X}_t)$, as explained in the proof of Proposition 3. Moreover, we have that

$$|R_2(\Xi_t)| \le \frac{L}{2} \|\Xi_t\|_F^2 \le L\delta_t^2 \tag{C.38}$$

In view of the above, we have:

$$\|\mathbb{E}[\hat{\mathbf{V}}_{i,t} \mid \mathcal{F}_t] - \mathbf{V}_i(\mathbf{X}_t)\|_F = D_i L \delta_t \tag{C.39}$$

and, therefore,

$$\|\mathbb{E}[\hat{\mathbf{V}}_t \mid \mathcal{F}_t] - \mathbf{V}(\mathbf{X}_t)\|_F = \sqrt{2} D L \delta_t \tag{C.40}$$

*(ii)* By the definition of $\hat{\mathbf{V}}_{i,t}$, we have:

$$\|\hat{\mathbf{V}}_{i,t}\|_F = \frac{D_i}{\delta_t} \left| u_i(\mathbf{X}_t^{(\delta)} + \delta_t \mathbf{Z}_t) \right| \|\mathbf{Z}_{i,t}\|_F \le \frac{D_i B}{\delta_t} \tag{C.41}$$

and therefore, we readily obtain that:

$$\mathbb{E}\left[\|\hat{\mathbf{V}}_{i,t}\|_F^2 \mid \mathcal{F}_t\right] \le \frac{D_i^2 B^2}{\delta_t^2} \tag{C.42}$$

We thus obtain

$$\mathbb{E}\left[\|\hat{\mathbf{V}}_t\|_F^2 \mid \mathcal{F}_t\right] \le \frac{2D^2 B^2}{\delta_t^2} \tag{C.43}$$

and our proof is complete. ∎

The only step missing is the proof of the actual guarantee of (3MW) with bandit feedback. We restate and prove the relevant result below:

**Theorem 3.** *Suppose that each player of a 2-player zero-sum game $\mathcal{Q}$ follows* (3MW) *for $T$ epochs with learning rate $\gamma$, sampling radius $\delta$, and gradient estimates provided by* (1PE)*. Then the players' empirical frequency of play enjoys the duality gap guarantee*

$$\mathbb{E}\left[\text{Gap}_{\mathcal{L}}(\bar{\mathbf{X}}_T)\right] \le \frac{H}{\gamma T} + \frac{2D^2 B^2 \gamma}{\delta^2} + 16DL\delta \tag{21}$$

*where $H = \log(d_1 d_2)$. In particular, for $\gamma = \left(\frac{H}{2T}\right)^{3/4} \frac{1}{2D\sqrt{BL}}$ and $\delta = \left(\frac{H}{2T}\right)^{1/4} \sqrt{\frac{B}{4L}}$, the players enjoy the equilibrium convergence guarantee:*

$$\mathbb{E}\left[\text{Gap}_{\mathcal{L}}(\bar{\mathbf{X}}_T)\right] \le \frac{2^{3/4} 8 H^{1/4} D \sqrt{BL}}{T^{1/4}}. \tag{22}$$

*Proof.* Following the same procedure as in the proof of Theorem 2, we readily obtain:

$$\begin{aligned}
\text{Gap}_{\mathcal{L}}(\bar{\mathbf{X}}_T) &\le \frac{1}{T} \sum_{t=1}^{T} \text{tr}[\mathbf{V}(\mathbf{X}_t)^\dagger (\mathbf{X}^* - \mathbf{X}_t)] \\
&\le \frac{1}{T} \sum_{t=1}^{T} F_t \left(\frac{1}{\gamma_t} - \frac{1}{\gamma_{t-1}}\right) + \frac{1}{2T} \sum_{t=1}^{T} \gamma_t \|\hat{\mathbf{V}}_t\|_F^2 \\
&\quad + \frac{4}{T} \sum_{t=1}^{T} \|b_t\|_F + \frac{1}{T} \sum_{t=1}^{T} \text{tr}[U_t^\dagger (\mathbf{X}_t - \mathbf{X}^*)]
\end{aligned} \tag{C.44}$$

Now, we have:

$$\mathbb{E}[\text{tr}[U_t^\dagger (\mathbf{X}_t - \mathbf{X}^*)]] = \mathbb{E}[\mathbb{E}[\text{tr}[U_t^\dagger (\mathbf{X}_t - \mathbf{X}^*)] \mid \mathcal{F}_t]] = 0 \tag{C.45}$$

since $\mathbf{X}_t$ is $\mathcal{F}_t$-measurable and $\mathbb{E}[U_t \mid \mathcal{F}_t] = 0$.

Moreover, by [Proposition 4](#), we have:

$$\|b_{i,t}\|_F = \left\|\mathbb{E}\big[\hat{\mathbf{V}}_{i,t} \mid \mathcal{F}_t\big] - \mathbf{V}_i(\mathbf{X}_t)\right\|_F \le 2D_i L\delta_t \tag{C.46}$$

and

$$\mathbb{E}\left[\left\|\hat{\mathbf{V}}_{i,t}\right\|_F^2\right] = \mathbb{E}\left[\mathbb{E}\left[\left\|\hat{\mathbf{V}}_{i,t}\right\|_F^2 \mid \mathcal{F}_t\right]\right] \le \frac{D_i^2 B^2}{\delta_t^2} \tag{C.47}$$

Hence, taking expectation in [(D.34)](#), we obtain:

$$\mathbb{E}\big[\mathrm{Gap}_{\mathcal{L}}(\bar{\mathbf{X}}_T)\big] \le \frac{1}{T}\sum_{t=1}^{T}\mathbb{E}[F_t]\left(\frac{1}{\gamma_t} - \frac{1}{\gamma_{t-1}}\right) + \frac{1}{2T}\sum_{t=1}^{T}\gamma_t\,\mathbb{E}[\|\hat{\mathbf{V}}_t\|_F^2] + \frac{4}{T}\sum_{t=1}^{T}\mathbb{E}[\|b_t\|_F]$$

$$\le \frac{1}{T}\sum_{t=1}^{T}\mathbb{E}[F_t]\left(\frac{1}{\gamma_t} - \frac{1}{\gamma_{t-1}}\right) + \frac{2D^2 B^2}{T}\sum_{t=1}^{T}\frac{\gamma_t}{\delta_t^2} + \frac{16DL}{T}\sum_{t=1}^{T}\delta_t \tag{C.48}$$

Setting $\gamma_t = \gamma$ and $\delta_t = \delta$, we obtain:

$$\mathbb{E}\big[\mathrm{Gap}_{\mathcal{L}}(\bar{\mathbf{X}}_T)\big] \le \frac{\mathbb{E}[F_1]}{\gamma T} + \frac{2D^2 B^2 \gamma}{\delta^2} + 16DL\delta \tag{C.49}$$

and finally, noting that

$$\mathbb{E}[F_1] = D(\mathbf{X}^*, \mathbf{X}_1) \le \log(d_1 d_2) \tag{C.50}$$

we get:

$$\mathbb{E}\big[\mathrm{Gap}_{\mathcal{L}}(\bar{\mathbf{X}}_T)\big] \le \frac{H}{\gamma T} + \frac{2D^2 B^2 \gamma}{\delta^2} + 16DL\delta \tag{C.51}$$

where $H = \log(d_1 d_2)$. Hence, after tuning $\gamma$ and $\delta$ to optimize this last expression, our result follows by setting $\gamma = \left(\frac{H}{2T}\right)^{3/4} \frac{1}{2D\sqrt{BL}}$ and $\delta = \left(\frac{H}{2T}\right)^{1/4}\sqrt{\frac{B}{4L}}$. $\blacksquare$

# D   Omitted proofs from [Section 6](#)

We provide first the bounds of the estimator [(1PE)](#) in a $N$-player quantum game. Formally, we have:

**Lemma D.1.** *The estimator* [(1PE)](#) *in a $N$-player quantum game $\mathcal{Q}$ enjoys the conditional bounds*

$$(i)\ \ \|\mathbb{E}[\hat{\mathbf{V}}_t \mid \mathcal{F}_t] - \mathbf{V}(\mathbf{X}_t)\|_F \le \frac{1}{2}DLN^{3/2}\delta_t \quad and \quad (ii)\ \ \mathbb{E}[\|\hat{\mathbf{V}}_t\|_F^2 \mid \mathcal{F}_t] \le \frac{D^2 B^2 N}{\delta_t^2}. \tag{D.1}$$

*Proof.* We prove each part separately.

*(i)* Let $\Xi_{i,t}$ be defined for all players $i \in \mathcal{N}$:

$$\Xi_{i,t} = \mathbf{X}_{i,t}^{(\delta)} - \mathbf{X}_{i,t} = \delta_t \mathbf{Z}_{i,t} + \frac{\delta_t}{r_i}(\mathbf{R}_i - \mathbf{X}_{i,t}) = \delta_t\left[\mathbf{Z}_{i,t} + \frac{1}{r_i}(\mathbf{R}_i - \mathbf{X}_{i,t})\right] \tag{D.2}$$

Taking a 1st-order Taylor expansion of $u_i$, we obtain:

$$u_i(\mathbf{X}_t^{(\delta)} + \delta_t \mathbf{Z}_t) = u_i(\mathbf{X}_t) + \sum_{j \in \mathcal{N}} \mathrm{tr}\left[\nabla_{\mathbf{X}_j^\top} u_i(\mathbf{X}_t)^\dagger \Xi_{j,t}\right] + R_2(\Xi_t) \tag{D.3}$$

Since $\mathbb{E}[U_i(\omega_t) \mid \mathcal{F}_t, \mathbf{Z}_t] = u(\mathbf{X}_t^{(\delta)} + \delta_t \mathbf{Z}_t)$, combining it with [(D.3)](#), we readily get:

$$\mathbb{E}[\hat{\mathbf{V}}_{i,t} \mid \mathcal{F}_t, \mathbf{Z}_t] = \frac{D_i}{\delta_t}u_i(\mathbf{X}_t^{(\delta)} + \delta_t \mathbf{Z}_t)\,\mathbf{Z}_{i,t}$$

$$= \frac{D_i}{\delta_t}u_i(\mathbf{X}_t)\mathbf{Z}_{i,t} + \frac{D_i}{\delta_t}\sum_{j \in \mathcal{N}}\mathrm{tr}\left[\nabla_{\mathbf{X}_j^\top} u_i(\mathbf{X}_t)^\dagger \Xi_{j,t}\right]\mathbf{Z}_{i,t} + \frac{D_i}{\delta_t}R_2(\Xi_t)\mathbf{Z}_{i,t} \tag{D.4}$$

Now, because $\mathbb{E}\big[\mathbf{Z}_{i,t}\,\big|\,\mathcal{F}_t\big] = 0$ and $\mathbf{Z}_{i,t}$ is sampled independent of any other process, we have:

$$\mathbb{E}\big[u_i(\mathbf{X}_t)\mathbf{Z}_{i,t}\,\big|\,\mathcal{F}_t\big] = u_i(\mathbf{X}_t)\,\mathbb{E}\big[\mathbf{Z}_{i,t}\,\big|\,\mathcal{F}_t\big] = 0 \tag{D.5}$$

and for $j \neq i \in \mathcal{N}$:

$$\mathbb{E}\Big[\mathrm{tr}\Big[\nabla_{\mathbf{X}_j^\top} u_i(\mathbf{X}_t)^\dagger \Xi_{j,t}\Big]\mathbf{Z}_{i,t}\,\Big|\,\mathcal{F}_t\Big] = 0 \tag{D.6}$$

Therefore, we obtain:

$$
\begin{aligned}
\mathbb{E}\left[\sum_{j\in\mathcal{N}} \mathrm{tr}\Big[\nabla_{\mathbf{X}_j^\top} u_i(\mathbf{X}_t)^\dagger \Xi_{j,t}\Big]\mathbf{Z}_{i,t}\;\Big|\;\mathcal{F}_t\right] &= \mathbb{E}\big[\mathrm{tr}\big[\mathbf{V}_i(\mathbf{X}_t)^\dagger \Xi_{i,t}\big]\mathbf{Z}_{i,t}\;\big|\;\mathcal{F}_t\big] \\
&= \delta_t\,\mathbb{E}\big[\mathrm{tr}\big[\mathbf{V}_i(\mathbf{X}_t)^\dagger \mathbf{Z}_{i,t}\big]\mathbf{Z}_{i,t}\;\big|\;\mathcal{F}_t\big] \\
&= \frac{\delta_t}{D_i}\sum_{W\in\mathcal{E}_i}\mathrm{tr}\big[\mathbf{V}_i(\mathbf{X}_t)^\dagger W\big]W \\
&= \frac{\delta_t}{D_i}\,\mathrm{proj}_{\mathcal{E}_i}(\mathbf{V}_i(\mathbf{X}_t)) = \frac{\delta_t}{D_i}\mathbf{V}_i(\mathbf{X}_t) \tag{D.7}
\end{aligned}
$$

where in the last step, we identify $\mathrm{proj}_{\mathcal{E}_i}(\mathbf{V}_i(\mathbf{X}_t))$ with $\mathbf{V}_i(\mathbf{X}_t)$, as explained in the proof of Proposition 3. Moreover, we have that

$$|R_2(\Xi_t)| \leq \frac{L}{2}\|\Xi_t\|_F^2 \leq \frac{1}{2}LN\delta_t^2 \tag{D.8}$$

In view of the above, we have:

$$\|\mathbb{E}[\hat{\mathbf{V}}_{i,t}\,|\,\mathcal{F}_t] - \mathbf{V}_i(\mathbf{X}_t)\|_F \leq \frac{1}{2}D_i LN\delta_t \tag{D.9}$$

and, therefore,

$$\|\mathbb{E}[\hat{\mathbf{V}}_t\,|\,\mathcal{F}_t] - \mathbf{V}(\mathbf{X}_t)\|_F \leq \frac{1}{2}DLN^{3/2}\delta_t \tag{D.10}$$

*(ii)* By the definition of $\hat{\mathbf{V}}_{i,t}$, we have:

$$\|\hat{\mathbf{V}}_{i,t}\|_F = \frac{D_i}{\delta_t}\Big|u_i(\mathbf{X}_t^{(\delta)} + \delta_t\,\mathbf{Z}_t)\Big|\,\|\mathbf{Z}_{i,t}\|_F \leq \frac{D_i B}{\delta_t} \tag{D.11}$$

and therefore, we readily obtain that:

$$\mathbb{E}\big[\|\hat{\mathbf{V}}_{i,t}\|_F^2\,\big|\,\mathcal{F}_t\big] \leq \frac{D_i^2 B^2}{\delta_t^2} \tag{D.12}$$

Hence, ultimately, we get the bound

$$\mathbb{E}\big[\|\hat{\mathbf{V}}_t\|_F^2\,\big|\,\mathcal{F}_t\big] \leq \frac{D^2 B^2 N}{\delta_t^2} \tag{D.13}$$

and our proof is complete. ∎

With all this in hand, we are finally in a position to proceed with the proof of Theorem 4, which we restate below for convenience:

**Theorem 4.** *Fix some tolerance level $\eta \in (0,1)$ and suppose that the players of an $N$-player quantum game follow* (3MW) *with bandit, realization-based feedback, and surrogate gradients provided by the estimator* (1PE) *with step-size and sampling radius parameters such that*

$$(i)\ \sum_{t=1}^{\infty} \gamma_t = \infty, \quad (ii)\ \sum_{t=1}^{\infty} \gamma_t\delta_t < \infty, \quad and \quad (ii)\ \sum_{t=1}^{\infty} \gamma_t^2/\delta_t^2 < \infty. \tag{23}$$

*If $\mathbf{X}^*$ is variationally stable, there exists a neighborhoold $\mathcal{U}$ of $\mathbf{X}^*$ such that*

$$\mathbb{P}(\lim_{t\to\infty}\mathbf{X}_t = \mathbf{X}^*) \geq 1 - \eta \quad whenever\ \mathbf{X}_1 \in \mathcal{U}. \tag{24}$$

*Proof.* Since $\mathbf{X}^*$ is variationally stable, there exists a neighborhood $\mathcal{U}_{vs}$ of it such that

$$\mathrm{tr}[\mathbf{V}(\mathbf{X})(\mathbf{X} - \mathbf{X}^*)] < 0 \quad \text{for all } \mathbf{X} \in \mathcal{U}_{vs} \backslash \{\mathbf{X}^*\}. \tag{D.14}$$

For any $\varepsilon' > 0$, defining

$$\mathcal{U}'_\varepsilon := \{\mathbf{X} \in \mathcal{X} : D(\mathbf{X}^*, \mathbf{X}) < \varepsilon'\} \tag{D.15}$$

we readily obtain by the continuity of $\mathbf{X} \mapsto D(\mathbf{X}^*, \mathbf{X})$ at $\mathbf{X}^*$ that there exists a neighborhood $\mathcal{U}_\varepsilon$ of $\mathbf{X}^*$ such that $\mathcal{U}_\varepsilon \subseteq \mathcal{U}_{vs}$. Note that if $\varepsilon_1 < \varepsilon_2$, we automatically get that $\mathcal{U}_{\varepsilon_1} \subseteq \mathcal{U}_{\varepsilon_2}$.

In view of this, we let $\mathbf{X}_1 \in \mathcal{U}_{\varepsilon/4} \subseteq \mathcal{U}_\varepsilon \subseteq \mathcal{U}_{vs}$. We divide the rest of the proof in steps.

### Step 1. Deriving the general energy inequality

By [Lemma 1](#) we have that:

$$D(\mathbf{X}^*, \mathbf{X}_{t+1}) \le D(\mathbf{X}^*, \mathbf{X}_t) + \gamma_t \,\mathrm{tr}[\hat{\mathbf{V}}_t(\mathbf{X}_t - \mathbf{X}^*)] + \frac{\gamma_t^2}{2}\|\hat{\mathbf{V}}_t\|_F^2. \tag{D.16}$$

Decomposing $\hat{\mathbf{V}}_t$ into

$$\hat{\mathbf{V}}_t = \mathbf{V}(\mathbf{X}_t) + b_t + U_t \tag{D.17}$$

as per (C.20) and applying (D.16) inequality iteratively, we get that

$$
\begin{aligned}
D(\mathbf{X}^*, \mathbf{X}_{t+1}) &\le D(\mathbf{X}^*, \mathbf{X}_1) + \sum_{s=1}^t \gamma_s \,\mathrm{tr}[\hat{\mathbf{V}}_s(\mathbf{X}_s - \mathbf{X}^*)] + \frac{1}{2}\sum_{s=1}^t \gamma_s^2\|\hat{\mathbf{V}}_s\|_F^2 \\
&\le D(\mathbf{X}^*, \mathbf{X}_1) + \sum_{s=1}^t \gamma_s \,\mathrm{tr}[\mathbf{V}(\mathbf{X}_s)(\mathbf{X}_s - \mathbf{X}^*)] + \sum_{s=1}^t \gamma_s \,\mathrm{tr}[b_s(\mathbf{X}_s - \mathbf{X}^*)] \\
&\quad + \sum_{s=1}^t \gamma_s \,\mathrm{tr}[U_s(\mathbf{X}_s - \mathbf{X}^*)] + \frac{1}{2}\sum_{s=1}^t \gamma_s^2\|\hat{\mathbf{V}}_s\|_F^2 \\
&\le D(\mathbf{X}^*, \mathbf{X}_1) + \sum_{s=1}^t \gamma_s \,\mathrm{tr}[\mathbf{V}(\mathbf{X}_s)(\mathbf{X}_s - \mathbf{X}^*)] + \sum_{s=1}^t \gamma_s \|b_s\|_F\|\mathbf{X}_s - \mathbf{X}^*\|_F \\
&\quad + \sum_{s=1}^t \gamma_s \,\mathrm{tr}[U_s(\mathbf{X}_s - \mathbf{X}^*)] + \frac{1}{2}\sum_{s=1}^t \gamma_s^2\|\hat{\mathbf{V}}_s\|_F^2 \\
&\le D(\mathbf{X}^*, \mathbf{X}_1) + \sum_{s=1}^t \gamma_s \,\mathrm{tr}[\mathbf{V}(\mathbf{X}_s)(\mathbf{X}_s - \mathbf{X}^*)] + \mathrm{diam}(\mathcal{X})\sum_{s=1}^t \gamma_s\|b_s\|_F \\
&\quad + \sum_{s=1}^t \gamma_s \,\mathrm{tr}[U_s(\mathbf{X}_s - \mathbf{X}^*)] + \frac{1}{2}\sum_{s=1}^t \gamma_s^2\|\hat{\mathbf{V}}_s\|_F^2
\end{aligned}
\tag{D.18}
$$

Defining the processes $\Psi_t$, $M_t$ and $Z_t$ for $t = 1, 2, \ldots$ as

$$\Psi_t := \frac{1}{2}\sum_{s=1}^t \gamma_s^2\|\hat{\mathbf{V}}_s\|_F^2 \tag{D.19a}$$

$$M_t := \sum_{s=1}^t \gamma_s \,\mathrm{tr}[U_s(\mathbf{X}_s - \mathbf{X}^*)] \tag{D.19b}$$

$$Z_t := \mathrm{diam}(\mathcal{X})\sum_{s=1}^t \gamma_s\|b_s\|_F \tag{D.19c}$$

equation (D.18) can be rewritten as

$$D(\mathbf{X}^*, \mathbf{X}_{t+1}) \le D(\mathbf{X}^*, \mathbf{X}_1) + \sum_{s=1}^t \gamma_s \,\mathrm{tr}[\mathbf{V}(\mathbf{X}_s)(\mathbf{X}_s - \mathbf{X}^*)] + Z_t + M_t + \Psi_t \tag{D.20}$$

### Step 2. Bounding the noise terms

Let $\varepsilon > 0$ as defined in the beginning of the proof.

- Regarding the term $Z_t$, it is clear that the process $\{Z_t : t \geq 1\}$ is a sub-martingale. Hence, by Doob's maximal inequality for sub-martingales [24], we get that:

$$
\begin{aligned}
\mathbb{P}\left(\sup_{s \leq t} Z_s \geq \varepsilon/4\right) &\leq \frac{\mathbb{E}[Z_t]}{\varepsilon/4} \\
&\leq \frac{\operatorname{diam}(\boldsymbol{\mathcal{X}}) \sum_{s=1}^{t} \gamma_s \, \mathbb{E}[\|b_s\|_F]}{\varepsilon/4} \\
&\leq \frac{\operatorname{diam}(\boldsymbol{\mathcal{X}}) \sum_{t=1}^{\infty} \gamma_t \, \mathbb{E}[\|b_t\|_F]}{\varepsilon/4} \\
&= \frac{\operatorname{diam}(\boldsymbol{\mathcal{X}}) \sum_{t=1}^{\infty} \gamma_t \, \mathbb{E}[\mathbb{E}[\|b_t\|_F \mid \mathcal{F}_t]]}{\varepsilon/4} \\
&\leq \frac{2 \operatorname{diam}(\boldsymbol{\mathcal{X}}) D L N^{3/2} \sum_{t=1}^{\infty} \gamma_t \delta_t}{\varepsilon}
\end{aligned}
\tag{D.21}
$$

By ensuring that

$$
\sum_{t=1}^{\infty} \gamma_t \delta_t \leq \frac{\varepsilon \eta}{6 \operatorname{diam}(\boldsymbol{\mathcal{X}}) D L N^{3/2}}
\tag{D.22}
$$

and taking $t$ go to $\infty$, (D.21) becomes:

$$
\mathbb{P}\left(\sup_{t \geq 1} Z_t \geq \varepsilon/4\right) \leq \eta/3
\tag{D.23}
$$

- Similarly, it is clear that the process $\{\Psi_t : t \geq 1\}$ is a sub-martingale. Following the same procedure, by Doob's maximal inequality for sub-martingales [24], we get that:

$$
\begin{aligned}
\mathbb{P}\left(\sup_{s \leq t} \Psi_s \geq \varepsilon/4\right) &\leq \frac{\mathbb{E}[\Psi_t]}{\varepsilon/4} \leq \frac{\frac{1}{2} \sum_{s=1}^{t} \gamma_s^2 \, \mathbb{E}\left[\|\hat{\mathbf{V}}_s\|_F^2\right]}{\varepsilon/4} \\
&\leq \frac{\frac{1}{2} \sum_{t=1}^{\infty} \gamma_t^2 \, \mathbb{E}\left[\|\hat{\mathbf{V}}_t\|_F^2\right]}{\varepsilon/4} \\
&\leq \frac{2 D^2 B^2 N \sum_{t=1}^{\infty} \gamma_t^2/\delta_t^2}{\varepsilon}
\end{aligned}
\tag{D.24}
$$

By ensuring that

$$
\sum_{t=1}^{\infty} \gamma_t^2/\delta_t^2 \leq \frac{\varepsilon \eta}{6 D^2 B^2 N}
\tag{D.25}
$$

and taking $t \to \infty$, (D.24) becomes:

$$
\mathbb{P}\left(\sup_{t \geq 1} \Psi_t \geq \varepsilon/4\right) \leq \eta/3
\tag{D.26}
$$

- Finally, regarding the term $M_t$, the process $\{M_t : t \geq 1\}$ is a martingale. Following the same procedure, by Doob's maximal inequality for martingales [24], we get that:

$$
\begin{aligned}
\mathbb{P}\left(\sup_{s \leq t} M_s \geq \varepsilon/4\right) &\leq \mathbb{P}\left(\sup_{s \leq t} |M_s| \geq \varepsilon/4\right) \leq \frac{\mathbb{E}[M_t^2]}{(\varepsilon/4)^2} = \frac{\sum_{s=1}^{t} \gamma_s^2 \, \mathbb{E}\left[\operatorname{tr}[U_s(\mathbf{X}_s - \mathbf{X}^*)]^2\right]}{(\varepsilon/4)^2} \\
&\leq \frac{\operatorname{diam}(\boldsymbol{\mathcal{X}})^2 \sum_{s=1}^{t} \gamma_s^2 \, \mathbb{E}\left[\|U_s\|_F^2\right]}{(\varepsilon/4)^2} \\
&\leq \frac{\operatorname{diam}(\boldsymbol{\mathcal{X}})^2 \sum_{t=1}^{\infty} \gamma_t^2 \, \mathbb{E}\left[\|U_t\|_F^2\right]}{(\varepsilon/4)^2} \\
&\leq \frac{4 \operatorname{diam}(\boldsymbol{\mathcal{X}})^2 \sum_{t=1}^{\infty} \gamma_t^2 \, \mathbb{E}\left[\|\hat{\mathbf{V}}_t\|_F^2\right]}{(\varepsilon/4)^2} \\
&\leq \frac{4 \operatorname{diam}(\boldsymbol{\mathcal{X}})^2 D^2 B^2 N \sum_{t=1}^{\infty} \gamma_t^2/\delta_t^2}{(\varepsilon/4)^2}
\end{aligned}
\tag{D.27}
$$

where we used the fact that

$$
\mathbb{E}[M_t^2] = \mathbb{E}\left[ \sum_{s=1}^{t} \gamma_s^2 \mathrm{tr}[U_s(\mathbf{X}_s - \mathbf{X}^*)]^2 + \sum_{k<\ell} \gamma_k \gamma_\ell \, \mathrm{tr}[U_k(\mathbf{X}_k - \mathbf{X}^*)] \, \mathrm{tr}[U_\ell(\mathbf{X}_\ell - \mathbf{X}^*)] \right]
$$

$$
= \mathbb{E}\left[ \sum_{s=1}^{t} \gamma_s^2 \mathrm{tr}[U_s(\mathbf{X}_s - \mathbf{X}^*)]^2 \right] \tag{D.28}
$$

itself following from the total expectation

$$
\mathbb{E}[\mathrm{tr}[U_k(\mathbf{X}_k - \mathbf{X}^*)] \, \mathrm{tr}[U_\ell(\mathbf{X}_\ell - \mathbf{X}^*)]] = \mathbb{E}[\mathbb{E}[\mathrm{tr}[U_k(\mathbf{X}_k - \mathbf{X}^*)] \, \mathrm{tr}[U_\ell(\mathbf{X}_\ell - \mathbf{X}^*)] \mid \mathcal{F}_\ell]]
$$

$$
= \mathbb{E}[\mathrm{tr}[U_k(\mathbf{X}_k - \mathbf{X}^*)] \, \mathbb{E}[\mathrm{tr}[U_\ell(\mathbf{X}_\ell - \mathbf{X}^*)] \mid \mathcal{F}_\ell]]
$$

$$
= 0 \tag{D.29}
$$

Now, by ensuring that

$$
\sum_{t=1}^{\infty} \gamma_t^2/\delta_t^2 \leq \frac{(\varepsilon/4)^2 \eta}{12 \, \mathrm{diam}(\mathcal{X})^2 D^2 B^2 N} \tag{D.30}
$$

and taking $t$ go to $\infty$, (D.27) becomes:

$$
\mathbb{P}\left( \sup_{t \geq 1} M_t \geq \varepsilon/4 \right) \leq \eta/3 \tag{D.31}
$$

Therefore, combining (D.23), (D.26) and (D.31) and applying a union bound, we get:

$$
\mathbb{P}\left( \left\{ \sup_{t \geq 1} Z_t \geq \varepsilon/4 \right\} \cup \left\{ \sup_{t \geq 1} \Psi_t \geq \varepsilon/4 \right\} \cup \left\{ \sup_{t \geq 1} M_t \geq \varepsilon/4 \right\} \right) \leq \eta \tag{D.32}
$$

Thus, defining the event $E := \left\{ \sup_{t \geq 1} Z_t + \Psi_t + M_t < \frac{3}{4}\varepsilon \right\}$, Eq. (D.32) readily implies that:

$$
\mathbb{P}(E) \geq 1 - \eta \tag{D.33}
$$

**Step 3. $\mathbf{X}_t \in \mathcal{U}_{\mathrm{vs}}$ with high probability**

Since $\mathbf{X}_1 \in \mathcal{U}_{\varepsilon/4} \subseteq \mathcal{U}_{\mathrm{vs}}$, by induction on $t$ we have that under the event $E$

$$
D(\mathbf{X}^*, \mathbf{X}_{t+1}) \leq D(\mathbf{X}^*, \mathbf{X}_1) + \sum_{s=1}^{t} \gamma_s \, \mathrm{tr}[\mathbf{V}(\mathbf{X}_s)(\mathbf{X}_s - \mathbf{X}^*)] + Z_t + M_t + \Psi_t \tag{D.34}
$$

$$
\leq \frac{\varepsilon}{4} + \frac{\varepsilon}{4} + \frac{\varepsilon}{4} + \frac{\varepsilon}{4} = \varepsilon \tag{D.35}
$$

where in the last step we used the inductive hypothesis that $\mathbf{X}_s \in \mathcal{U}_{\mathrm{vs}}$ for all $s = 1, \dots, t$, which implies $\mathrm{tr}[\mathbf{V}(\mathbf{X}_s)(\mathbf{X}_s - \mathbf{X}^*)] < 0$. This implies that $\mathbf{X}_{t+1} \in \mathcal{U}_\varepsilon \subseteq \mathcal{U}_{\mathrm{vs}}$.

Therefore, we obtain that $\mathbf{X}_{t+1} \in \mathcal{U}_\varepsilon \subseteq \mathcal{U}_{\mathrm{vs}}$ for all $t \geq 1$. For the rest of the proof we will work under the event $E$.

**Step 4. Subsequential convergence**

Now we will show that there exists a subsequence $\{\mathbf{X}_{t_k} : k \geq 1\}$ suct that $\lim_{k \to \infty} \mathbf{X}_{t_k} = \mathbf{X}^*$. Suppose it does not. Then, this would mean that the quantity $\mathrm{tr}[\mathbf{V}(\mathbf{X}_t)(\mathbf{X}_t - \mathbf{X}^*)]$ is bounded away from zero. Combining it with the fact that $\mathbf{X}_t \in \mathcal{U}_{\mathrm{vs}}$ for all $t \geq 0$, we readily get that there exists $c > 0$ such that:

$$
\mathrm{tr}[\mathbf{V}(\mathbf{X}_s)(\mathbf{X}_s - \mathbf{X}^*)] < -c \tag{D.36}
$$

Then, (D.20) would give:

$$
D(\mathbf{X}^*, \mathbf{X}_{t+1}) \leq \varepsilon - c \sum_{s=1}^{t} \gamma_s \tag{D.37}
$$

Hence, taking $t \to \infty$, and using that $\sum_{t \geq 1} \gamma_t = \infty$, we would get that $D(\mathbf{X}^*, \mathbf{X}_t) \to -\infty$, which is a contradiction, since $D(\mathbf{X}^*, \mathbf{X}_t) \geq 0$.

Hence, there exists a subsequence $\{\mathbf{X}_{t_k} : k \geq 1\}$ suct that $\lim_{k \to \infty} \mathbf{X}_{t_k} = \mathbf{X}^*$, i.e.,

$$
\lim_{k \to \infty} D(\mathbf{X}^*, \mathbf{X}_{t_k}) = 0. \tag{D.38}
$$

**Step 5. Existence of** $\lim_{t\to\infty} D(\mathbf{X}^*, \mathbf{X}_t)$

We define the sequence of events $\{E_t : t \geq 1\}$ as

$$E_t := \left\{ \sup_{s \leq t-1} Z_s + \Psi_s + M_s < \frac{3}{4}\varepsilon \right\} \quad \text{for } t \geq 2 \tag{D.39}$$

and

$$E_1 := \{\mathbf{X}_1 \in \mathcal{U}_{\varepsilon/4}\} \tag{D.40}$$

Then we have that $E_t \in \mathcal{F}_t$ and $E_t \subseteq \{\mathbf{X}_s \in \mathcal{U}_{\mathrm{vs}} : s = 1, \ldots, t\}$.

Defining the random process $\{\tilde{D}_t : t \geq 1\}$ as

$$\tilde{D}_t = D(\mathbf{X}^*, \mathbf{X}_t) \, \mathbb{1}_{E_t} \tag{D.41}$$

Then, by (D.16) we have

$$D(\mathbf{X}^*, \mathbf{X}_{t+1}) \leq D(\mathbf{X}^*, \mathbf{X}_1) + \gamma_t \operatorname{tr}[\mathbf{V}(\mathbf{X}_t)(\mathbf{X}_t - \mathbf{X}^*)] + \operatorname{diam}(\mathcal{X})\gamma_t \|b_t\|_F$$
$$+ \gamma_t \operatorname{tr}[U_t(\mathbf{X}_t - \mathbf{X}^*)] + \frac{1}{2}\gamma_t^2 \|\hat{\mathbf{V}}_t\|_F^2 \tag{D.42}$$

Multiplying the above relation with $\mathbb{1}_{E_t}$, and noting that $\mathbb{1}_{E_{t+1}} \leq \mathbb{1}_{E_t}$, since $E_{t+1} \subseteq E_t$, we have

$$\tilde{D}_{t+1} \leq \tilde{D}_t + \gamma_t \operatorname{tr}[\mathbf{V}(\mathbf{X}_t)(\mathbf{X}_t - \mathbf{X}^*)] \, \mathbb{1}_{E_t} + \operatorname{diam}(\mathcal{X})\gamma_t \|b_t\|_F \, \mathbb{1}_{E_t}$$
$$+ \gamma_t \operatorname{tr}[U_t(\mathbf{X}_t - \mathbf{X}^*)] \, \mathbb{1}_{E_t} + \frac{1}{2}\gamma_t^2 \|\hat{\mathbf{V}}_t\|_F^2 \, \mathbb{1}_{E_t} \tag{D.43}$$

$$\leq \tilde{D}_t + \operatorname{diam}(\mathcal{X})\gamma_t \|b_t\|_F \, \mathbb{1}_{E_t} + \gamma_t \operatorname{tr}[U_t(\mathbf{X}_t - \mathbf{X}^*)] \, \mathbb{1}_{E_t} + \frac{1}{2}\gamma_t^2 \|\hat{\mathbf{V}}_t\|_F^2 \, \mathbb{1}_{E_t} \tag{D.44}$$

where in the last step we used that $\operatorname{tr}[\mathbf{V}(\mathbf{X}_t)(\mathbf{X}_t - \mathbf{X}^*)] \, \mathbb{1}_{E_t} \leq 0$. Therefore, we obtain that:

$$\mathbb{E}[\tilde{D}_{t+1} \mid \mathcal{F}_t] \leq \tilde{D}_t + \operatorname{diam}(\mathcal{X})\gamma_t \, \mathbb{1}_{E_t} \, \mathbb{E}[\|b_t\|_F \mid \mathcal{F}_t] + \frac{1}{2}\gamma_t^2 \, \mathbb{1}_{E_t} \, \mathbb{E}[\|\hat{\mathbf{V}}_t\|_F^2 \mid \mathcal{F}_t] \tag{D.45}$$

where we used that

$$\mathbb{E}\big[\operatorname{tr}[\mathbf{V}(\mathbf{X}_t)(\mathbf{X}_t - \mathbf{X}^*)] \, \mathbb{1}_{E_t} \, \big| \, \mathcal{F}_t\big] = \mathbb{1}_{E_t} \, \mathbb{E}[\operatorname{tr}[\mathbf{V}(\mathbf{X}_t)(\mathbf{X}_t - \mathbf{X}^*)] \mid \mathcal{F}_t] = 0 \tag{D.46}$$

Therefore, $\{\tilde{D}_t : t \geq 1\}$ is an almost super-martingale [52] and, thus, there exists $\tilde{D}_\infty$ with $\tilde{D}_\infty$ finite (a.s.) and $\tilde{D}_t \to \tilde{D}_\infty$ (a.s.).

Since $E = \cap_{t \geq 1} E_t$, we have:

$$\mathbb{P}\Big(\lim_{t\to\infty} D(\mathbf{X}^*, \mathbf{X}_t) \text{ exists} \, \Big| \, E\Big) = \frac{\mathbb{P}(\{\lim_{t\to\infty} D(\mathbf{X}^*, \mathbf{X}_t) \text{ exists}\} \cap E)}{\mathbb{P}(E)} \tag{D.47}$$

$$= \frac{\mathbb{P}\big(\{\lim_{t\to\infty} \tilde{D}_t \text{ exists}\} \cap E\big)}{\mathbb{P}(E)} = 1 \tag{D.48}$$

Hence, $\lim_{t\to\infty} \tilde{D}_t$ exists on $E$ and by *Step 3* we readily get that $\lim_{t\to\infty} \tilde{D}_t = 0$ on $E$. Thus, by Lemma A.2, we get

$$\lim_{t\to\infty} \mathbf{X}_t = \mathbf{X}^* \quad \text{on the event } E \tag{D.49}$$

and setting $\mathcal{U} = \mathcal{U}_{\varepsilon/4}$, we obtain

$$\mathbb{P}\Big(\lim_{t\to\infty} \mathbf{X}_t = \mathbf{X}^*\Big) \geq 1 - \eta \quad \text{whenever } \mathbf{X}_1 \in \mathcal{U}. \tag{D.50}$$

This concludes our discussion and our proof. ∎

# E  Numerical experiments

In this last appendix, we provide a series of additional numerical simulations to validate and explore the performance of (MMW) with payoff-based feedback.

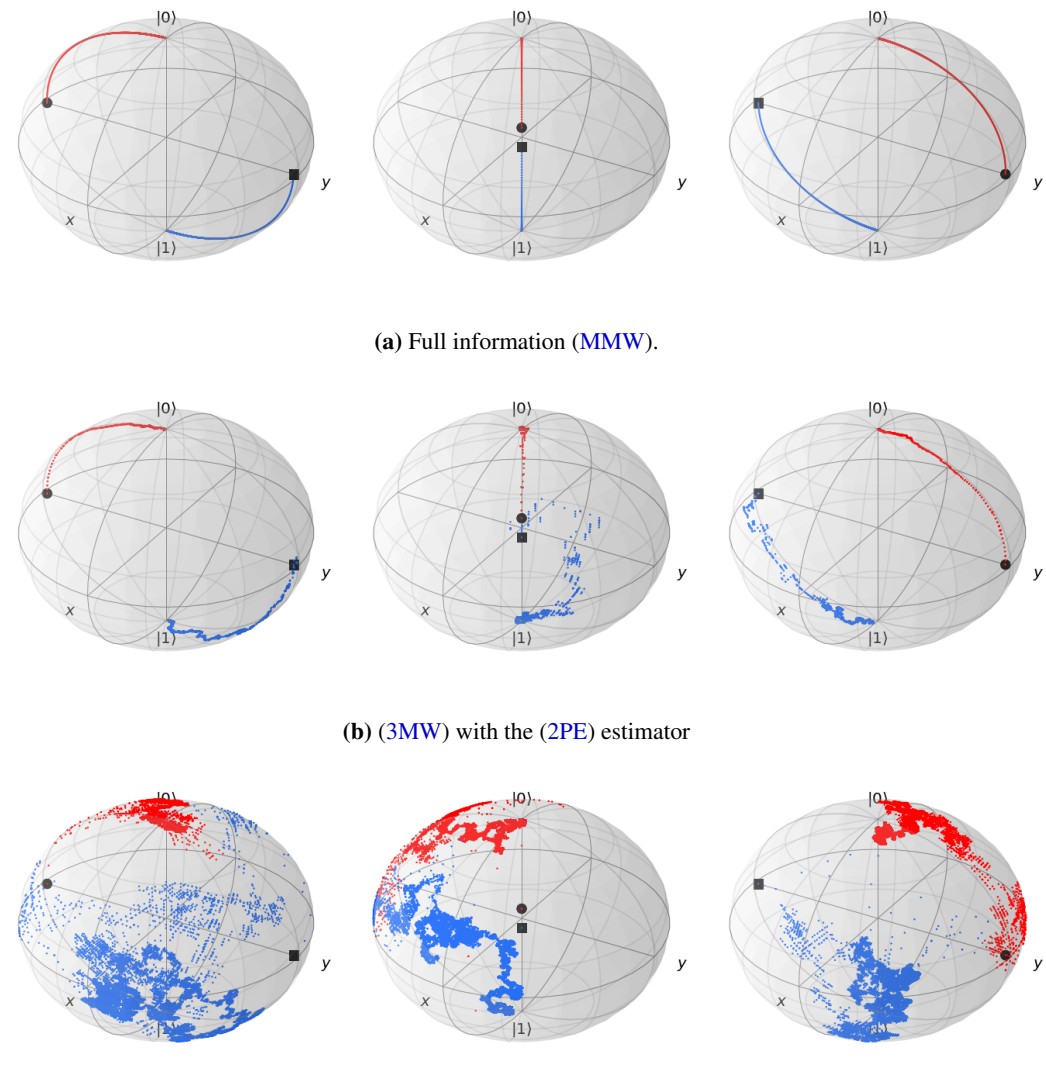

**(a)** Full information (MMW).

**(b)** (3MW) with the (2PE) estimator

**(c)** (3MW) with the (1PE) estimator

**Figure 2:** Trajectories of the three methods for different initial conditions. The red points correspond to player 1, and the blue points to player 2. The initial points of the red trajectories are marked with •, while the initial points of the blue ones are marked with ■.

**Trajectory analysis.**   First, we proceed to a trajectory analysis of the game setup presented in Section 7. Specifically, in Fig. 2, we provide a visualization of the actual trajectories of play generated by the three methods with the same parameters as before, for different initial conditions. The trajectories are presented in Bloch spheres [48], where the points $|0\rangle$ and $|1\rangle$ in the figure correspond to the density matrices

$$|0\rangle = \begin{pmatrix} 1 & 0 \\ 0 & 0 \end{pmatrix} \qquad \text{and} \qquad |1\rangle = \begin{pmatrix} 0 & 0 \\ 0 & 1 \end{pmatrix} \tag{E.1}$$

respectively. In all figures, the points in red indicate the trajectory of Player 1, while the points in blue are for Player 2. The initial points of the red trajectories are marked with •, while the initial points of the blue ones are marked with ■. [Each column of Bloch spheres in Fig. 2 has the same initial conditions.]

An important remark here is that, as suggested by Theorem 4, the trajectories of all methods converge – and quite rapidly at that – to the game's (strict) Nash equilibrium. In fact, given that the trajectories converge to a pure state, this goes to explain the faster convergence rates observed in Fig. 1: instead

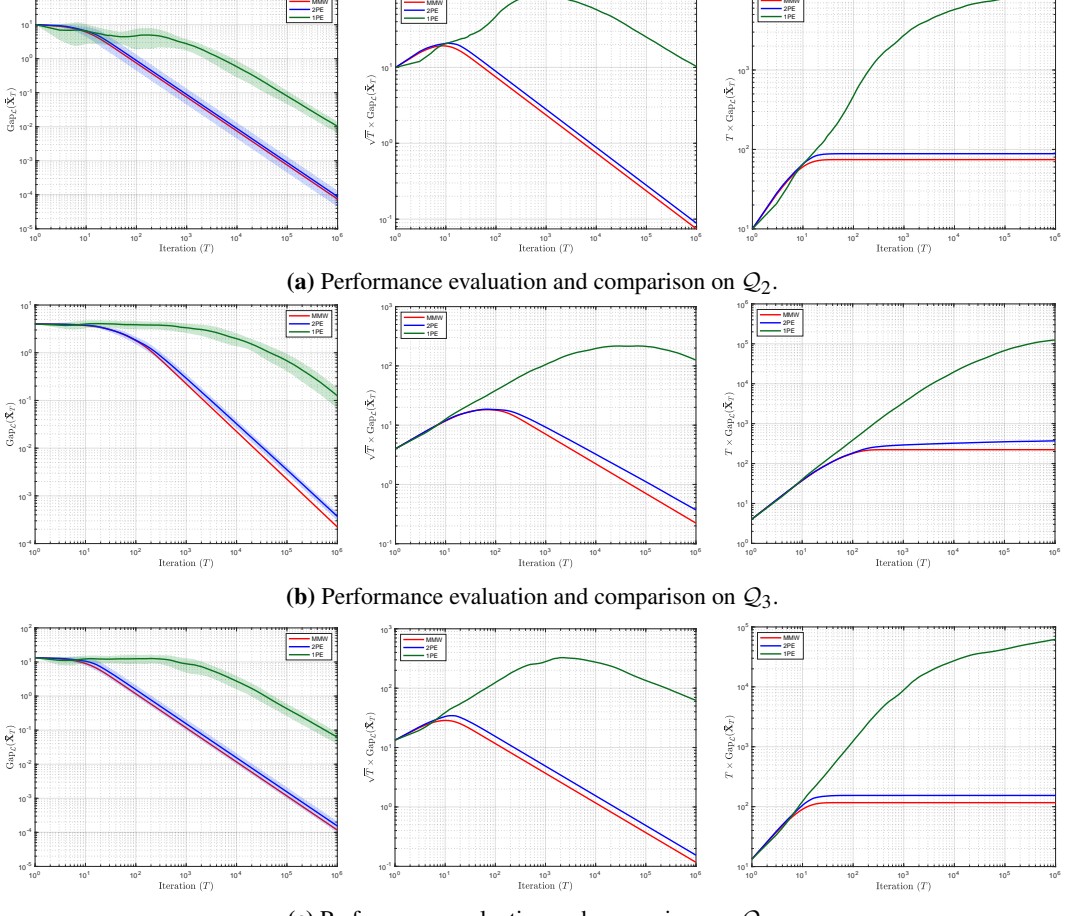

**(a)** Performance evaluation and comparison on $\mathcal{Q}_2$.

**(b)** Performance evaluation and comparison on $\mathcal{Q}_3$.

**(c)** Performance evaluation and comparison on $\mathcal{Q}_4$.

**Figure 3:** Performance evaluation of (3MW) with estimators provided by (2PE) and (1PE), and comparison with the full information algorithm (MMW).

of oscillating around a solution, the MMW orbits actually *converge* to equilibrium in this case, so the trailing average converges at a much faster rate. This holds in all zero-sum games with a pure equilibrium, thus indicating a very important class of zero-sum games where the worst-case guarantees of MMW algorithms can be significantly improved.

**Convergence speed analysis.** In addition to the game setup described in Section 7, we consider the following quantum games:

- $\mathcal{Q}_2$: quantum analogue of the $2 \times 2$ min-max game with payoff matrix

$$P_2 = \begin{pmatrix} (10, -10) & (10, -10) \\ (-10, 10) & (-10, 10) \end{pmatrix} \tag{$\mathcal{Q}_2$}$$

- $\mathcal{Q}_3$: quantum analogue of the $3 \times 3$ min-max game with payoff

$$P_3 = \begin{pmatrix} (4, -4) & (2, -2) & (4, -4) \\ (-4, 4) & (-2, 2) & (-4, -4) \\ (-4, 4) & (-2, 2) & (-4, -4) \end{pmatrix} \tag{$\mathcal{Q}_3$}$$

- $\mathcal{Q}_4$: quantum analogue of the $3 \times 3$ min-max game with payoff

$$P_4 = \begin{pmatrix} (10, -10) & (10, -10) & (10, -10) \\ (-10, 10) & (-10, 10) & (-10, 10) \\ (-10, 10) & (-10, 10) & (-10, 10) \end{pmatrix} \tag{$\mathcal{Q}_4$}$$

In Fig. 3, we evaluate the convergence properties of (3MW) using the estimators (2PE) and (1PE), and compare it with the full information variant (MMW), following the same setup as described in Section 7. Specifically, for each method, we perform 10 different runs, with $T = 10^5$ steps each, and compute the mean value of the duality gap as a function of the iteration $t = 1, 2, \ldots, T$. The solid lines correspond to the mean values of the duality gap of each method, and the shaded regions enclose the area of $\pm 1$ (sample) standard deviation among the 10 different runs. Note that the red line, which corresponds to the full information (MMW), does not have a shaded region, since there is no randomness in the algorithm. All the runs for the three different methods were initialized for $\mathbf{Y} = 0$ and we used $\gamma = 10^{-2}$ for all methods. In particular, for (3MW) with gradient estimates given by (2PE) estimator, we used a sampling radius $\delta = 10^{-2}$, and for (3MW) with (1PE) estimator, we used $\delta = 10^{-1}$ (in tune with our theoretical results which suggest the use of a tighter sampling radius when mixed payoff information is available to the players). As highlighted in the main text, we observe that the decrease in performance is mild, and the different algorithms achieved better rates than their theoretical guarantees.

