# OpenReview forum: "Payoff-based Learning with Matrix Multiplicative Weights in Quantum Games"
_NeurIPS.cc/2023/Conference — NeurIPS 2023 poster_

### Official Review · Reviewer_UZra · 2023-07-03

**Soundness:** 3 good
**Presentation:** 2 fair
**Contribution:** 3 good
**Rating:** 7
**Confidence:** 4

**Summary:**

The paper studies the setting of learning in quantum games, in the case where agents only have access to a scalar (for instance, a payoff value) instead of their full individual gradient, and have to update their strategies in order to converge to some equilibrium. The primary contribution is a variant of the well-studied MMW algorithm called 3MW, which matches the equilibrium convergence rate of (full information) MMW in 2-player quantum zero-sum games. Moreover, with a relaxation of 3MW’s information requirements (i.e. that players observe a random realization of their payoff observable), the algorithm converges to equilibrium at O(T^{-1/4}) rate. Finally, in general N-player quantum games, a regularized version of 3MW using estimator 1PE is able to locally converge to Nash equilibria that are variationally stable.

**Strengths:**

The paper answers an important question in the realm of learning in quantum games - in this framework it is quite unreasonable to assume full information of gradients, especially without properly defining how measurements are made, how players access said measurements etc. Hence, in my opinion the bandit framework is very fitting for the quantum setting, and indeed players should be able to update their strategies in a truly ‘online’ setting using only their measured/realized payoff observable. The techniques used in the paper, while not entirely novel, show interesting similarities and differences between classical and quantum settings. Finally, I think the results on variational stability are quite significant, as not only does this give (to my knowledge) the broadest last-iterate convergence result in N-player quantum games yet, it also points to a potential class of FTRL variants using bandit feedback that could potentially also be convergent in the last-iterate sense.

**Weaknesses:**

I think overall this is a strong paper, the only issue to me is that most of the techniques and ideas used in the paper are not particularly different from the classical bandit setting. If there is significant technical novelty, it is not made clear to the reader in the main text. I would be very interested to see what the differences are in the analysis between the quantum and classical setting, if there are any.

**Questions:**

In the experiments, 3MW is compared with MMW and is shown to exhibit decent performance even with the lack of information. Does the practical performance of 3MW deteriorate significantly for games of larger dimension? To what extent can we rely on 3MW in practical settings (i.e. large games, games with larger number of players)?

**Limitations:**

Limitations are very briefly discussed, and perhaps a longer discussion would improve the paper (if space allows)

---

> ### Author Rebuttal · Authors · 2023-08-09
>
> Thank you for your encouraging input and strongly positive evaluation. To streamline our response, we reply to your questions and comments in a point-by-point thread below, and we will revise our manuscript accordingly at the next revision opportunity:
>
> 1. "*In the experiments, 3MW is compared with MMW and is shown to exhibit decent performance even with the lack of information. Does the practical performance of 3MW deteriorate significantly for games of larger dimension? To what extent can we rely on 3MW in practical settings (i.e. large games, games with larger number of players)?*"
>
>     **Reply.**  In the current context, the goal of our simulations was to illustrate the behavior of our algorithm in simple examples, providing some additional intuition and corroborating the theoretical results. However, we believe that the performance of the algorithm would remain analogous in larger games. An extensive study of the performance of 3MW in specific game-theoretic scenarios (especially with more players) would be a very fruitful direction, and one that we would seek to undertake in future research to better understand the performance envelope of MMW and 3MW in larger multi-agent systems.
>
> ---
>
> 2. "*Most of the techniques and ideas used in the paper are not particularly different from the classical bandit setting. If there is significant technical novelty, it is not made clear to the reader in the main text. I would be very interested to see what the differences are in the analysis between the quantum and classical setting.*"
>
>    **Reply.** First, we would like to point out that in classical, finite games, the de facto way of learning with bandit-type feedback is via the use of IWE techniques. However, as we explain in Section 4.1, this is not possible in our case because of the quantum / semidefinite nature of the underlying game. More precisely, in a classical game, the set of pure strategies is finite, so any mixed strategy is purely atomic by default. By contrast, the set of pure strategies in quantum games is continuous (a Hilbert sphere), so a generic probability distribution over this set would be nonatomic. The IWE approach only makes sense for purely atomic distributions (because it requires dividing by the probability with which a specific pure strategy was employed), so it cannot be employed in a setting with a continuum of pure actions. In addition, because of the specific setup of quantum games, mixing over this continuum of pure states does not lead to a (continuous) probability distribution, but to a density matrix, so the use of standard bandit techniques based on an IWE approach is made even less relevant.
>
>    As for the specific technical challenges that arise in the matrix setting, most obstacles concerned the design of the Hermitian sampling frame that underlies the definition of estimators 2PE and 1PE (cf. Propositions 1 and 2), and the derivation of the associated bounds for each estimator (Propositions 3 and 4). The reason that this particular frame was required was because the standard zeroth-order sampling scheme of Flaxman et al. [17] requires drawing a sampling direction uniformly at random from a sphere. In our case however, we also had to respect the eigenvalue constraints of the spectraplex and sample from a specific tangent space (by contrast, the original paper of Flaxman et al. did not encounter this complication because it only considered sampling from convex bodies, which the spectraplex is not). To circumvent this obstacle, we had to design a Hermitian sampling frame and an associated feasibility correction term that would preserve the structure of the spectraplex (cf. Appendix B). However, because any sampling frame can only cover a finite number of directions, the sampled function was not "smoothed out" (as in the case of Flaxman et al.), so the established zeroth-order approximation arguments could not be employed in our case. This then had to be paired with the lengthy calculations involved in Propositions 3 and 4 (cf. Appendix C) that were needed to derive the anytime convergence guarantees of the 2PE and 1PE schemes in Theorems 2 and 3 respectively.
>
> ---
>
> We hope and trust that these points address your questions - thank you again for your encouraging remarks and positive evaluation!
>
> Kind regards,
>
> The authors

---

> > ### Comment · Reviewer_UZra · 2023-08-15
> > **Response to Author Rebuttal**
> >
> > Thank you for your answers to the questions. The explanation for differences between classical and quantum proofs for the bandit setting is quite compelling, and it would be nice to see a subsection summarizing this in the main text if space allows. The experiments added above are also quite helpful for a practical comparison. Finally, I find making it clear that the paper is about classical algorithms for quantum games to be quite imperative - it can be easy to conflate the two settings.
> >
> > Best regards,
> > Reviewer UZra

---

> > > ### Author Response · Authors · 2023-08-21
> > >
> > > Thank you very much for your input and positive evaluation. We will update our manuscript accordingly.
> > >
> > >
> > > Regards,
> > >
> > > The authors

---

### Official Review · Reviewer_Kb52 · 2023-07-04

**Soundness:** 3 good
**Presentation:** 3 good
**Contribution:** 2 fair
**Rating:** 6
**Confidence:** 5

**Summary:**

This paper studies online learning of quantum algorithms based on the matrix multiplicative weight (MMW) updates. A main technical contribution is to introduce a weaker version called the minimal-information matrix multiplicative weights (3MW), which only requires zeroth-order gradient sampler and can achieve an overall regret of O(T^{-1/4}). In addition, for N-player quantum games, the paper proved a convergence result that 3MW guarantees local convergence with high probability to all equilibria that satisfy a certain first-order stability condition.

**Strengths:**

From my perspective, the most notable strength of this paper is the nice observation that less information is required to make MMW works. Considering that MMW has wide applications in quantum algorithms for machine learning and optimization, I believe that the result may be of general interest. Technically, this is achieved by exploiting ideas from machine learning, in particular bandit optimization, which provides interesting perspective for designing future quantum algorithms.

**Weaknesses:**

The most notable weakness is the claim in intro that the result by Jain and Watrous [21] with O(\sqrt{T}) regret for quantum online learning algorithms remains the tightest known bound for Nash equilibrium learning in quantum games. This is not true, as a recent paper by Gao et al. https://arxiv.org/abs/2304.14197 gave a quantum algorithm with poly-loga regret for zero-sum games. Compared to this paper, the results obtained by this paper is sub-optimal.

Another notable weakness is that the dependence in dimension is at least D, and does not present a quantum speedup here. In fact, a main merit of applying MMW into quantum algorithms is that MMW provides fast quantum algorithms for solving relevant problems. An important paper was Brandao and Svore https://arxiv.org/pdf/1609.05537.pdf in FOCS 2017, which can solve semidefinite programs with complexity square-root in D. If the current framework 3MW has at least linear dependence in D, it will prohibit its further applications in quantum computing.

**Questions:**

1. Can the authors further comment on whether the convergence rate of the proposed 3MW algorithm, or in other words the online regret, can be further improved? Given the existence of works such as Gao et al. mentioned above, it would be helpful if the authors can provide a more complete picture here.

2. I’m confused by the sentence “in contrast to classical finite games, quantum games have an infinite continuum of pure states” in the abstract. On classical computers, even if we play games with finite cardinality, we can still choose mixed strategies and the set of probability distributions supported on a finite set can be uncountable.

3. It would in general be helpful if the authors can introduce more on the applications of 3MW. In particular, in MMW the payoff gradient of player i is defined as V_i(X_t). However, in most existing quantum papers, V_i is very simple: the identity function. For instance, in the above Brandao-Svore paper as well as several important follow-up works that solves semidefinite programs with quantum speedup, V_i is the identity function and the update is simply a learning rate \eta * X_t.

In all, it’s nice to know that 3MW can be applied with bandit information and for general V_i, but this should probably be further motivated about why we need to consider more complicated V_i’s.

4. In numerical experiments, around Page 29, the authors claimed that the their algorithms achieved a rate of convergence closer O(1/T) instead of the weaker O(1/\sqrt{T}) or O(1/T^{1/4}) bounds. However, I didn’t see detailed evidence for this; it would be helpful if the authors can present more detailed data and plots for demonstrating this phenomenon.

Note that this is related to my question 1: if the convergence rate is ~O(1/T), the overall online regret will be poly-log. So it is probably worth closer study about this, both theoretically and experimentally.

**Limitations:**

This work is mainly theoretical so I don't think there are notable limitations, but it would work better if the authors can articulate this more clearly.

---

> ### Author Rebuttal · Authors · 2023-08-09
>
> Thank you for your input and positive evaluation. We reply to your questions and comments point-by-point below:
>
> 1. "*[The $O(\sqrt{T})$ regret of Jain & Watrous is not the tightest known bound.] A recent paper by Gao et al. gave a quantum algorithm with poly-log regret for zero-sum games.*"
>
>    Thanks for bringing this work to our attention. This paper was posted on arxiv just three weeks before the NeurIPS submission deadline so we were not aware of it at the time, but we will be happy to discuss it in detail in our revision.
>
>    First, we would like to point out that Gao et al. employ a **quantum algorithm to solve classical, finite zero-sum games**. By contrast, our paper (as well as Jain & Watrous) study **classical computing algorithms for solving quantum games.** These two questions are mutually orthogonal, and it is not possible to compare results across papers: in particular, **Gao et al. do not offer any guarantees for quantum zero-sum games;** their analysis focuses exclusively on the (optimistic) multiplicative weights algorithm for classical, finite zero-sum games, so it does not apply to our context.
>
>    The quantum speedup of Gao et al. is due to a quantum Gibbs sampling oracle that relies on coherent quantum access to the matrix of the game and the use of quantum registers and quantum RAM storage. In our setting however, we no longer have probability distributions living in the **simplex** but density matrices living in the **spectraplex**, so the quantum sampling oracle cannot be applied. Augmenting the MMW algorithm via the use of quantum computing would be a fruitful direction for future research, but one which lies beyond the scope of our work.
>
> ---
> 2. "*The dependence in dimension is at least D, and does not present a quantum speedup. [...] Brandao and Svore (2017) can solve semidefinite programs with complexity square-root in D.*"
>
>    As above, Brandao and Svore propose a **quantum** algorithm for **classical** semidefinite programs. By contrast, we employ *classical* computing to solve **quantum** games, so an improvement in $D$ would clash with the $\Omega(D/\sqrt{T})$ lower bound of Dani et al., *The price of bandit information for online optimization* (NeurIPS 2008). Enhancing MMW via quantum computing would be a fruitful direction for future research, but one which lies beyond the scope of our work.
>
> ---
> 3. "*I'm confused by the sentence [regarding pure states] in the abstract.*"
>
>    The relevant phrase reads "*quantum games have an infinite continuum of pure states [...] so standard importance-weighting techniques for estimating payoff vectors cannot be employed.*" This is explained in Section 4.1, where we detail the IWE approach for bandit learning in finite games.
>
>    In short, the main difficulty is as follows: in classical games, the set of pure strategies is finite, so mixed strategies are purely atomic. By contrast, the set of pure states in quantum games is a Hilbert sphere, so a generic probability distribution over this set would be nonatomic. The IWE approach only makes sense for purely atomic distributions (because it divides by pure strategy probabilities), so it cannot be employed against a continuum of pure actions. Also, in quantum games, we do not have probability distributions but *density matrices*, so the use of IWE techniques is even less relevant.
>
> ---
> 4. "*In most existing quantum papers, V_i is the identity function.*"
>
>    It is not true that $V_i$ is the identity matrix in general - this is not even the case in classical games (where $V_i$ is diagonal, but not the identity).
>
>    In more detail, the objective function of a quantum zero-sum game is of the form $$L(X,Y) = \sum _{i,j=1}^{d_1} \sum _{k,\ell=1}^{d_2} X _{ij} Y _{k\ell} W _{ijk\ell}$$ where $W$ is the game's payoff bi-array (cf. "Semidefinite Games" by Ickstadt et al). This gives $V _{ij}^{(1)} = \sum _{k,\ell=1}^{d_2} W _{ijk\ell} Y _{k\ell}$ for Player 1 (and analogously for Player 2) so, unless $W$ has a very special form (which is not the case even in quantum games coming from *classical* games), there is no reason for $V$ to be the identity matrix (or diagonal).
>
> ---
> 5. "*It would be helpful if the authors can say more on the applications of 3MW.*"
>
>    Just like MMW, 3MW can be applied to online kernel learning [45], covariance matrix optimization in vector Gaussian channels [Telatar, 1999; Yu, Rhee, Boyd & Cioffi, 2004], etc. For example, in the latter case, we have the player-action structure of a quantum game but with payoffs $$
>    u_i(X) = \log\det\left(I + \sum_j H_j X_j H_j^\dagger\right) $$ where $H_i$ is the gain matrix of player $i$, and each $X_i$ lies on the spectraplex. Since $\log\det$ is concave, our analysis extends directly to this setting by replacing the quantum game gap function in (C.23) and (C.24) with the above.  [We also note that $V_i$ is not the identity here either.]
>
>    We did not insist on these further applications because we did not want to dilute the focus of our paper, but we would be happy to take advantage of the extra page to detail all this.
> ---
> 6. "*The authors [observed] a rate closer to O(1/T) [...] it would be helpful if the authors can present more detailed data.*"
>
>    We performed additional experiments to illustrate this behavior in the one-page rebuttal PDF. In more detail, we plotted $T \mathrm{Gap}(T)$ and $\sqrt{T} \mathrm{Gap}(T)$ to estimate the rate of convergence of the duality gap. Our experiments show that both 2PE and 1PE stabilize very close to $1/T$, and are faster than $1/\sqrt{T}$. This is what motivated our comment that the convergence rate in our experiments is closer to $O(1/T)$ and is due to the fact that the sequence of iterates of MMW/3MW converges itself, not only in time-averages (in line with our analysis in Section 6); we will expand further on this in the final version.
>
> ---
> Please let us know if you have any further questions - and thank you again for your input and positive evaluation.

---

> > ### Comment · Reviewer_Kb52 · 2023-08-10
> > **Acknowledgement**
> >
> > I would like to thank the authors for the very detailed reply and the additional numerical experiments. Those provide much clearer understandings, and I'm happy to subsequently increase my score.
> >
> > For the final version, I suggest the authors to: 1) highlight that the setting of this paper is to study classical algorithms for solving quantum games, and make comparison to quantum computing literature in other settings; 2) add the new numerical experiments; and 3) add discussions to my other questions for more clear presentation.

---

> > > ### Author Response · Authors · 2023-08-12
> > > **Thank you for your acknowledgment**
> > >
> > > Thank you for your quick reply and for upgrading your score, we will update our maunscript according to your detailed suggestions - thanks again for your helpful comments!
> > >
> > > Regards,
> > >
> > > The authors

---

### Official Review · Reviewer_G5vn · 2023-07-16

**Soundness:** 3 good
**Presentation:** 3 good
**Contribution:** 3 good
**Rating:** 6
**Confidence:** 3

**Summary:**

This paper studies Quantum Games (mainly in the two-player zero-sum setting) and develops payoff-based learning algorithms with provable convergence-rate guarantees. The learning algorithm combines matrix multiplicative weights (MMW) and zeroth order method for gradient estimation. The overall convergence rate is either $O(1/\sqrt{T})$ or $1/T^{1/4}$, depending the information structure. The authors also provide numerical simulations in the appendix to verify the performance of the proposed algorithm.

**Strengths:**

This paper is very well-written. The appendix is also of high quality.

**Weaknesses:**

(1) The learning algorithm combines MMW with zeroth order method for gradient estimation. Since the convergence rate of MMW was established in the literature, and the zeroth order method is well studied in optimization, the technical novelty is not entirely clear. Are there any significant challenges in the analysis?

(2) From Theorem 2, it seems that there are no consequences for choosing $\delta$ to be arbitrarily small. Am I missing anything?

**Questions:**

Do the authors expect the bound to be tight? Are there any known lower bounds for the convergence rate in the literature?

**Limitations:**

The authors did not explicitly point out the limitations of this work.

---

> ### Author Rebuttal · Authors · 2023-08-09
>
> Thank your for your input and positive evaluation. To streamline our response, we reply to your questions and comments in a point-by-point thread below, and we will revise our manuscript accordingly at the next revision opportunity:
>
> 1. *Do the authors expect the bound to be tight? Are there any known lower bounds for the convergence rate in the literature?*
>
>    In the context of convex-concave min-max problems without access to perfect gradient information, the associated lower bound is $\Omega(1/\sqrt{T})$ [Ouyang and Xu, *Lower complexity bounds of first-order methods for convex-concave bilinear saddle-point problems*, Mathematical Programming, 2021]. In this regard, the $O(1/\sqrt{T})$ guarantee of the 2PE scheme is order-optimal and cannot be tightened further.
>
>    The case of noisy function evaluations for 1PE is trickier. For convex **minimization** problems (a simpler class than min-max problems), the best-known lower bound is $\Omega(D/\sqrt{T})$ [Dani et al., *The price of bandit information for online optimization*, NeurIPS 2008]. As far as this lower bound is concerned, the guarantee (21) of Theorem 3 is tight in terms of $D$, but not in terms of $T$. To the best of our knowledge, the only implementable algorithm with an $O(1/\sqrt{T})$ convergence rate under comparable assumptions is the method of Bubeck et al. [*Kernel-based methods for bandit convex optimization*, STOC 2017]. However, this method has a much worse $\Theta(D^{9.5})$ dependence on the dimension which makes it impractical for most applications.
>
>    Importantly, the above bounds of Dani et al. and Bubeck et al. both concern **minimization** problems. The min-max case is considerably more complex and much less is known in that case, so we cannot tell whether the bound of Theorem 3 for the 1PE method is tight. We speculate that the dependence on $T$ may be improved via the use of a suitable semidefinite sampling kernel, but this is not clear, and even if true, such a modification would likely come at a cost of a much worse dependence on the dimensionality of the problem.
>
>    We were constrained by space in our original submission, but we would be happy to use the extra page of the revision to include the above discussion.
>
> ---
>
> 2. "*The learning algorithm combines MMW with zeroth order method for gradient estimation. Are there any significant challenges in the analysis?*"
>
>    First, we would like to point out that the MMW algorithm was conceived as a deterministic algorithm with full access to perfect gradient information. In that sense, it was not a given that it could be combined with a zeroth-order gradient estimator and remain convergent. (As an example to the contrary, Nesterov's accelerated gradient algorithm fails to converge altogether if run with stochastic gradients or zeroth-order estimates)
>
>    Regarding the specific technical challenges that arise in the matrix setting, most obstacles concerned the design of the Hermitian sampling frame that underlies the definition of estimators 2PE and 1PE (cf. Propositions 1 and 2), and the derivation of the associated bounds for each estimator (Propositions 3 and 4). The reason that this particular frame was required was because the standard zeroth-order sampling scheme of Flaxman et al. [17] involves drawing a sampling direction uniformly at random from a sphere. In our case however, we also had to respect the eigenvalue constraints of the spectraplex and sample from a specific tangent sphere (by contrast, the original paper of Flaxman et al. did not encounter this complication because it only considered sampling from convex bodies, which the spectraplex is not). To circumvent this obstacle, we had to design a Hermitian sampling frame and an associated feasibility correction term that would preserve the structure of the spectraplex (cf. Appendix B). However, because any sampling frame can only cover a finite number of directions, the sampled function was not "smoothed out" (as in the case of Flaxman et al.), so the established zeroth-order approximation arguments could not be employed in our case. This then had to be paired with the lengthy calculations involved in Propositions 3 and 4 (cf. Appendix C) that were needed to derive the anytime convergence rates of the 2PE and 1PE schemes in Theorems 2 and 3.
>
>    Finally, we should also point out that, to the best of our knowledge, there are no comparable results to the sequential convergence analysis of Section 6 for general games, even in the case of MMW with *full* gradient information - at least, we are not aware of a comparable analysis in the literature.
>
> ---
>
> 3. "*From Theorem 2, it seems that there are no consequences for choosing $\delta$ to be arbitrarily small. Am I missing anything?*"
>
>    This is true. The reason for this is that 2PE is run with mixed payoff observations, so the associated finite difference quotient is uniformly bounded. As a result, the variance of 2PE is always $O(1)$ and its bias can be made arbitrarily small by taking $\delta \to 0$, so there is no bias-variance trade-off in (14). The proposed value $\delta = (G/L) \sqrt{H/(8T)}$ was only intended to provide a compact expression for the algorithm's end guarantee; in theory, $\delta$ can be taken as small as the machine precision allows.
>
>    That said, any measurable error in the observation of the players' mixed payoffs would lead to finite differences that are $O(1)$, even when the sampled points are $O(\delta)$-close. In this case, 2PE would lead to a bias-variance trade-off similar to that of 1PE, and there would be a distinct disadvantage in taking $\delta$ arbitrarily small. Thus, in practice, $\delta$ would be limited by the machine precision of the players' calculating device and the associated floating-point accuracy.
>
> ---
>
> Please let us know if you have any further questions - and thank you again for your input and positive evaluation.
>
> Kind regards,
>
> The authors

---

> > ### Comment · Reviewer_G5vn · 2023-08-13
> > **Acknowledgement of the Rebuttal**
> >
> > Thank the authors for the detailed response. I do not have further questions.

---

> > > ### Author Response · Authors · 2023-08-21
> > >
> > > We are very pleased for answering your questions. Thank you again for your helpful comments and positive evaluation.
> > >
> > > Regards,
> > >
> > > The authors

---

### Official Review · Reviewer_TyoP · 2023-07-22

**Soundness:** 3 good
**Presentation:** 3 good
**Contribution:** 3 good
**Rating:** 5
**Confidence:** 3

**Summary:**

The authors consider the quantum games in the bandit setting, which players observe only the payoff value of their actions. In order to facilitate this, the authors extend the online bandit estimation methodology into the quantum-state space. As a result, they are able to realize a duality gap of $\mathcal O(T^{-\frac{1}{4}})$ and $\mathcal O(T^{-\frac{1}{2}})$ for bilinear quantum games.

**Strengths:**

- Originality & Significance: Considering the concept of games within quantum state spaces in the bandit setting is innovative, and the result is nice and matches those found in classical matrix games.

- Quality & Clarity: The paper is effectively structured, allowing readers to follow the presented arguments and conclusions with ease.

**Weaknesses:**

- The applications of blinear quantum games are questionable.

**Questions:**

- I would like to inquire about the necessity of the $L$-smooth assumption in the quantum game setting. Specifically, it seems that in classical games or online learning problems, the bandit estimator can be constructed without requiring an $L$-smooth condition. Could the authors elaborate on whether this condition is necessary in the context of quantum games and, if so, could you explain the rationale behind this requirement?

---

> ### Author Rebuttal · Authors · 2023-08-09
>
> Thank you for your encouraging input and positive evaluation. To streamline our response, we reply to your questions and comments in a point-by-point thread below, and we will revise our manuscript accordingly at the next revision opportunity:
>
> ---
>
> 1. "*In classical games or online learning problems, the bandit estimator can be constructed without requiring an $L$-smooth condition. Could the authors elaborate on whether this condition is necessary in the context of quantum games and, if so, could you explain the rationale behind this requirement?*"
>
>    **Reply.** When choosing the sampling direction uniformly at random from the unit sphere - e.g., as in the original paper of Flaxman et al. [17] - Lipschitz smoothness is not required because the resulting estimator is an unbiased stochastic gradient of a suitably "smoothed" version of the sampled function. In our case however, uniform sampling from a sphere introduces several computational complications because we need to also respect the eigenvalue constraints of the spectraplex and sample from a specific tangent sphere (by contrast, the original paper of Flaxman et al. only considered sampling from convex bodies, which the spectraplex is not).
>
>    These complications are not insurmountable, but we found it was computationally much more efficient to use the specific Hermitian frame that we designed in Proposition 1 to pick a sampling direction. When working with this particular sampling frame, Lipschitz smoothness appears in the conditional bounds of Proposition 3, and we are not aware of a technique to get rid of it (essentially because the sampling frame only covers a finite number of directions so it cannot "smooth out" the sampled function on its own). However, since quantum games are Lipschtiz smooth by default (thanks to the compactness of the spectraplex and the multilinearity of the players' payoff functions), the smoothness requirement comes "for free"; because of this, we leveraged the computational efficiency of the frame derived in Proposition 1.
>
> ---
>
> 2. "*The applications of blinear quantum games are questionable.*"
>
>    **Reply.** Some prominent applications of quantum games involve quantum generative adversarial networks (Q-GANs) [references 11,14,31 in our paper], (robust) shadow state tomography [refs. 1,2 in our paper], the best separable state (BSS) problem [see e.g., L. Gurvits, *Classical deterministic complexity of Edmonds’ problem and quantum entanglement*, STOC 2003], and many others. Admittedly, some of these applications involve quantum games that are not necessarily multilinear, but the theory of online learning in quantum games is still at its early stages, and the class of multilinear quantum games is a natural starting point.
>
>    At the same time - and perhaps more importantly - it should be noted that our paper provides an **application-agnostic** template for the analysis of the MMW algorithm with bandit information. In particular, our paper's results extend directly to other applications of the MMW algorithm, most notably to online kernel learning [the original motivation of Tsuda et al., ref. 45 in our paper], covariance matrix optimization in vector Gaussian channels [Capacity of multi-antenna Gaussian channels, Telatar, 1999; Yu et al., Iterative water-filling for Gaussian vector multiple-access channels, Transactions of Information Theory, 2004], etc. For example, in the latter case, the game being played has the same player-action structure as a quantum game but with payoff functions
>    $$
>    \textstyle
>    u_i(\mathbf{X}) = \log\det\left(\mathbf{I} + \sum_j \mathbf{H}_j \mathbf{X_j} \mathbf{H}_j^\dagger\right) - \log\det\left(\mathbf{I} + \sum _{j\neq i} \mathbf{H}_j \mathbf{X_j} \mathbf{H}_j^\dagger\right)
>    $$
>    where $\mathbf{H}_i$ is the gain matrix of player $i=1,\dotsc,N$, and each $\mathbf{X}_i$ is constrained to lie on the spectraplex $\mathbf{X}_i \succcurlyeq 0$, $\mathrm{tr}(\mathbf{X}_i) = 1$ (for a detailed formulation, see the paper by Yu et al. above). By the concavity of the $\log\det$ function, our analysis extends directly to this setting, simply by replacing the quantum game gap function in Eqs. (C.23) and (C.24) of the appendix with the aforementioned payoff structure.
>
>    We did not highlight these further applications of our approach because we were constrained by space and we did not want to dilute the focus of our paper. However, we would be happy to take advantage of the extra page provided in the revision phase to detail all this.
>
>
> ---
>
> We hope and trust that these points address your questions - but please let us know if any of the above is not sufficiently clear.
>
> Thank you again for your input and positive evaluation,
>
> The authors

---

> > ### Comment · Reviewer_TyoP · 2023-08-16
> >
> > I appreciate the authors addressing my concerns regarding the L-smoothness condition and the potential applications of bilinear quantum games. I would like to maintain the score within the acceptance threshold.

---

> > > ### Author Response · Authors · 2023-08-21
> > >
> > > We are very glad for addressing your questions. Thank you again for your remarks and your positive evaluation.
> > >
> > > Regards,
> > >
> > > The authors

---

### Author Rebuttal · Authors · 2023-08-09

Dear AC, dear reviewers,

We are sincerely grateful for your time, input and positive evaluation. To streamline the discussion phase, we reply to each reviewer's questions in a separate point-by-point thread below. We only include in this global rebuttal a pdf with some additional experiments to illustrate the convergence of the 3MW framework in different games (including a more precise estimation of its convergence rate).

We defer all other points to the reviewer-specific threads below and we are looking forward to the discussion phase if any further questions remain.

Kind regards,

The authors

---

### Decision · Program_Chairs · 2023-09-21

**Decision:**

Accept (poster)

**Comment:**

This paper studies online learning of quantum algorithms based on the matrix multiplicative weight (MMW) updates. A main technical contribution is to introduce a weaker version called the minimal-information matrix multiplicative weights (3MW), which only requires zeroth-order gradient sampler and can achieve an overall sublinear regret. In addition, for N-player quantum games, the paper proved a convergence result that 3MW guarantees local convergence with high probability to all equilibria that satisfy a certain first-order stability condition.

All reviewers believe this paper makes valid contributions to the community. The AC agrees and recommends acceptance.